# Histological and molecular characterization of the digestive system of early weaned juveniles of *Arapaima* sp. reared in a recirculating aquaculture system

Maria J. Darias[1]*, Guillain Estivals[2], Karl B. Andree[3], Christian Fernández-Méndez[4], Roger Bazán[5], Chantal Cahu[6], Enric Gisbert[3], Diana Castro-Ruiz[2]*

1 MARBEC, Univ Montpellier, CNRS, Ifremer, IRD, Montpellier, France, 2 Laboratorio de Biología y Genética Molecular (LBGM), Dirección de Investigación en Ecosistemas Acuáticos Amazónicos (AQUAREC), Instituto de Investigaciones de la Amazonía Peruana (IIAP), Iquitos, Peru, 3 Aquaculture Program, Institut de Recerca i Tecnología Agroalimentaries (IRTA), Sant Carles de la Ràpita, Spain, 4 Laboratorio de Bromatología, Dirección de Investigación en Ecosistemas Acuáticos Amazónicos (AQUAREC), Instituto de Investigaciones de la Amazonía Peruana (IIAP), Iquitos, Peru, 5 Dirección de Investigación en Ecosistemas Acuáticos Amazónicos (AQUAREC), Instituto de Investigaciones de la Amazonía Peruana (IIAP), Pucallpa, Peru, 6 LEMAR, Ifremer, Univ Brest, CNRS, IRD, Plouzané, France

* maria.darias@ird.fr (MJD); dcastro@iiap.gob.pe (DC-R)

## Abstract

*Arapaima* spp., the world's largest scaled freshwater fish, native to the Amazon and Essequibo river basins, are valued in aquaculture for their rapid growth and ornamental appeal. However, reliance on natural breeding and parental care in earthen ponds results in variable fingerling survival, hindering production. This study investigated the morphology and functionality of the digestive system of *Arapaima* sp. fingerlings from the Peruvian Amazon and evaluated the feasibility of early weaning onto compound diets to optimise growth and survival. Fingerlings were collected from a pond at 3.19±0.03 cm total length (TL) and reared in a recirculating aquaculture system at 29 ℃ under a 12L:12D photoperiod. Fish were successfully weaned from *Artemia* spp. nauplii onto an experimental compound diet (60% protein, 15% lipid) from 3.26±0.02 cm TL within three days. Histological and gene expression analyses of key digestive enzyme precursors and appetite-regulating peptides (*α-amylase*, *phospholipase A2*, *lipoprotein lipase*, *trypsinogen*, *chymotrypsinogen*, *pepsinogen*, and *peptide YY*) revealed a mature digestive system, with enhanced digestive efficiency observed at 5.05±0.34 cm TL. Based on digestive enzyme expression profiles and gut morphology, early juvenile *Arapaima* sp. possess a digestive physiology consistent with that of an omnivorous species with a preference for animal prey. The middle intestine was identified as a key site for fatty acid absorption and feed intake regulation. This study presents a novel, comprehensive analysis of digestive enzyme gene expression and associated tissue morphology in the genus *Arapaima*. It provides new insights into their digestive physiology and establishes the feasibility of early weaning

**Data availability statement:** All relevant data are within the paper and its Supporting Information files.

**Funding:** This research was funded by the Peruvian Project 192-FINCyT-IA-201 (Programa Nacional de Innovación para la Competitividad y Productividad, Innóvate-Perú, Peru) and the International Joint Laboratory 'Evolution and Domestication of the Amazonian Ichthyofauna' (LMI EDIA, IRD-IIAP-UAGRM, France, Peru, and Bolivia). DC-R received mobility grants from FONDECYT (Fondo Nacional de Desarrollo Científico, Tecnológico y de Innovación Tecnológica, Concurso Movilización Nacional e Internacional en Ciencia, Tecnología e Innovación (CTeI) 2014/2015, 013-2015-FONDECYT, Peru) and from the IRD (Mobilité Sud-Nord, France). The funders had no role in study design, data collection and analysis, decision to publish, or preparation of the manuscript.

**Competing interests:** The authors have declared that no competing interests exist.

onto formulated diets. Future research should explore the interplay between optimised compound feed formulations and refined early rearing protocols to maximise growth and survival throughout development.

## Introduction

Native to the Amazon and Essequibo river basins in South America, *Arapaima* spp. (Osteoglossidae) are the world's largest scaled freshwater fish, capable of reaching a maximum weight of 200 kg and a total length (TL) of 2–3 m [1,2]. Historically a major fishery resource, they hold significant aquaculture potential due to their rapid growth (reaching 70–100 cm in length and 10 kg in weight within the first year) and ornamental value [3,4]. *Arapaima* spp. populations have drastically declined due to overfishing, leading to their listing under CITES (Convention on International Trade in Endangered Species of Wild Fauna and Flora) Appendix II and classification as 'Data Deficient' on the IUCN Red List [3,5]. Traditionally considered a monotypic genus comprising solely *Arapaima gigas* (Schinz 1822), recent literature suggests the existence of multiple species or evolutionarily significant units within this genus [3,6–8], highlighting the importance of considering broodstock origin in aquaculture research and practices.

Initially classified as carnivorous [2,9], *Arapaima* spp. exhibit high dietary diversity and have been suggested to be secondary consumers with omnivorous feeding behaviours. However, uncertainty remains as to whether plant materials in their stomachs are intentionally ingested or incidental to feeding on animal prey [10,11]. Stomach content analysis shows that juveniles prefer insects, microcrustaceans, and gastropods, whereas adults primarily consume fish from lower trophic positions [11–13].

A key challenge in *Arapaima* spp. aquaculture is variable fingerling survival. Natural breeding occurs in earthen ponds, where parental care is provided for approximately three months after hatching. As obligate air breathers, they surface every 5–15 minutes to inhale air, with fingerlings surfacing for the first time 4–6 days post-hatching (dph), measuring 1.7 to 1.8 cm TL, and beginning to feed by 6 dph [3,4,14]. Fingerlings are typically harvested from the ponds between one and three weeks post-emersion (1.7 to 4 cm TL) and reared under controlled conditions to enhance survival rates [4]. While extensive research has focused on the feeding of larger juvenile *Arapaima* spp. (15–2000 g) [15–25], information on digestive capacities and nutritional requirements during early life stages (< 4 cm TL) remains scarce [26,27]. This knowledge gap hinders the development of optimised feeding protocols critical for maximising fingerling survival and growth, particularly for early weaning strategies.

Therefore, the present study investigated the digestive system morphology and functionality of *Arapaima* sp. fingerlings from the Peruvian Amazon and explored the potential for early weaning onto compound diets in a recirculating aquaculture system. Specifically, we examined histological characteristics of digestive tissues and the expression of a set of key digestive enzyme precursors and appetite-regulating

peptides [*α-amylase* (*amy*), *phospholipase A2* (*plA2*), *lipoprotein lipase* (*lpl*), *trypsinogen* (*try*), *chymotrypsinogen* (*ctr*), *pepsinogen* (*pga*), and *peptide YY* (*pyy*)]. This research aims to provide a fundamental understanding of early digestive development in *Arapaima* spp., informing the development of improved feeding and rearing strategies for enhanced aquaculture production.

## Materials and methods

### Rearing and feeding protocols

*Arapaima* sp. individuals were obtained from the natural reproduction of a breeding pair maintained in captivity in an earthen pond at the Instituto de Investigaciones de la Amazonía Peruana (IIAP, Iquitos, Peru). Given the taxonomic uncertainties within the genus *Arapaima* and the unclear distribution of its potential species [3,6–8], we refer to our study species from the Peruvian Amazon as *Arapaima* sp. Offspring were collected from the water surface of the pond at $3.19 \pm 0.03$ cm TL and transferred to six 30-L water volume tanks ($n = 6$) connected to a clear-water recirculating aquaculture system (initial density = 1 ind L$^{-1}$). Rearing conditions were maintained as follows: temperature, $29.0 \pm 0.03$ °C; pH, $7.97 \pm 0.05$; dissolved oxygen, $7.2 \pm 0.5$ mg L$^{-1}$; and a 12L:12D photoperiod. The experiment began after a two-day acclimatisation period under the new rearing conditions. During this period and on day 1 of the experiment (D1, $3.19 \pm 0.03$ cm TL), individuals were fed *Artemia* spp. nauplii five times daily (*ad libitum*). From D2 ($3.26 \pm 0.02$ cm TL), they were gradually weaned over three days onto an experimental compound diet formulated and manufactured at the Ifremer facilities (Brest, France; 60% crude protein and 15% lipid, expressed as a percentage of dry matter, particle size: 100–500 µm; Table 1). From day D5 ($3.43 \pm 0.01$ cm TL) onwards, the specimens were exclusively fed the compound diet.

**Table 1. Composition of the experimental diet used to wean *Arapaima* sp. at $3.26 \pm 0.02$ cm TL (D2 of culture). DM, dry matter.**

| Ingredients[1] (in % DM) | Experimental compound diet |
|---|---|
| Fishmeal | 62 |
| Hydrolyzed fishmeal (CPSP) | 17 |
| Lipids (marine lecithin) | 7 |
| Wheat starch | 7 |
| Vitamin mix [2] (x4) | 2 |
| Mineral mix [3] | 4 |
| Betain | 1 |
| *Analysis of the diet (% DM)* | |
| Proteins | 60 |
| Lipids | 15 |
| Carbohydrates | 6 |
| Energy (KJ) | 1653 |

[1]All dietary ingredients were commercially obtained. Fishmeal hydrolysate CPSP 90:10% lipids; soluble fish protein concentrate (Sopropêche, Boulogne sur Mer, France); marine lecithin LC 60 (Phosphotech, St Herblain, France).

[2]Composition per kilogram of vitamin mixture: choline chloride 60%, 333 g; vitamin A acetate, (4000 IU g$^{-1}$) 2 g; vitamin D3 (1920 IU g$^{-1}$) 0.96 g; vitamin E (40 IU g$^{-1}$) 20 g; vitamin B3 2 g, vitamin B5 4 g; vitamin B1 200 mg; vitamin B2 80%, 1 g; vitamin B6 600 mg; vitamin B9 80%, 250 mg; vitamin concentrate B12 (10 g kg$^{-1}$), 0.2 g; biotin, 1.5 g; vitamin K3. 51%, 3.92 g; meso-inositol 60 g; cellulose, 543.3 g.

[3]Composition per kilogram of mineral mixture: 90 g KCl, 40 mg KIO$_3$, 500 g CaHPO$_4$ 2H$_2$O, 40 g NaCl, 3 g CuSO$_4$ 5H$_2$O, 4 g ZnSO$_4$ 7H$_2$O, 20 mg CoSO$_4$ 7H$_2$O, 20 g FeSO$_4$ 7H$_2$O, 3 g MnSO$_4$ H$_2$O, 215 g CaCO$_3$, 124 g MgSO$_4$ 7H$_2$O, and 1 g NaF.

## Sampling and growth measurements

Fish were euthanised with an overdose of eugenol (0.05 µl ml$^{-1}$; Moyco®, Moyco, Lima, Peru). For growth measurements, groups of 4–14 individuals were sampled daily from D1 to D5, every other day from D5 to D11, and every three days from D11 to D17 (end of the trial). Specimens were photographed with a scale bar in a petri dish, and TL was measured on the images using ImageJ software [28]. Wet weight (WW) was measured using an analytical microbalance (Sartorius BP 211 D, Data Weighing Systems, Inc., Illinois, USA). Samples for histological and gene expression analyses were collected at D1 (3.19 ± 0.03 cm TL), D3 (3.31 ± 0.01 cm TL), D5 (3.43 ± 0.01 cm TL), D10 (4.00 ± 0.2 cm TL), and D15 (5.05 ± 0.34 cm TL). To include the earliest possible ontogenetic stage for the histological analyses, smaller individuals (2.03 ± 0.07 and 2.43 ± 0.07 cm TL) fed *Artemia* spp. nauplii *ad libitum* were collected from a separate offspring batch at the IIAP facilities in Pucallpa, Peru. Because fish of the same chronological age can exhibit substantial variation in digestive system development [29], development in *Arapaima* sp. was assessed based on TL.

Animal experimental procedures were carried out in compliance with the Guidelines of the European Union Council (2010/63/EU) regarding the protection of animals used for scientific purposes, as no *ad hoc* ethical committee was available at IIAP (Peru), where this study was conducted.

## Histological analyses

At each sampling point, 15 fish were fixed in buffered formaldehyde (pH 7.4) at 4 °C overnight, dehydrated the following day using a graded series of ethanol (50% and 70%), and stored in 70% ethanol at 4 °C until further processing. The fixed fish samples were then further dehydrated in 95% and 100% ethanol and embedded in paraffin using an automatic tissue processor (Histolab ZX-60Myr; Especialidades Médicas MYR SL, Tarragona, Spain). Paraffin blocks were prepared in an AP280–2Myr station and cut into serial sagittal sections (3 µm thick) using an automatic microtome Microm HM (Leica Microsystems Nussloch GmbH, Nussloch, Germany) [30]. The paraffin sections were kept at 40 °C overnight. Deparaffinisation was performed using a graded series of xylene, and the samples were stained with haematoxylin and eosin and trichromic VOF (light green, orange G, and acid fuchsin) stains [31] for general histomorphological observations. Periodic acid–Schiff (PAS) and Alcian Blue (AB) at pH 2.5 were used to detect neutral, carboxyl-rich, and sulphated glycoconjugates in mucous cells [32]. Observations and photographs of the histological preparations were made using a Leica DMLB microscope equipped with a digital camera Olympus DP70 (Olympus España, S.A.U., Barcelona, Spain). The numbers of (i) mucosal folds and goblet cells in the oesophagus, (ii) gastric glands in the stomach, (iii) lipid vacuoles in the liver and intestine, and (iv) hepatocytes and pancreatic cells were counted in six randomly selected fields per specimen (100 µm or 100 µm$^2$ per field, as applicable). Additionally, the size of (i) mucosal folds and goblet cells in the oesophagus and intestine, (ii) gastric glands in the stomach, (iii) the epithelium of the stomach and intestine, (iv) muscular layers of the stomach, and (v) lipid droplets in the liver and intestine were measured (in µm or µm$^2$, as appropriate) in six randomly selected fields per specimen [30]. The surface area of lipid droplets was calculated for 30 lipid droplets from five fish per sampling point and tank using the formula S = ¼ π $a$ $b$, where $a$ and $b$ are the minimum and maximum diameters, respectively [33]. ImageJ software was used for measurements on histological slides.

## Partial mRNA amplification and identification

For gene expression analyses, 200 mg WW of fish tissues (stomach, anterior, middle, and posterior intestines, and liver) were sampled from each tank per sampling day and preserved in RNAlater (Sigma-Aldrich, St. Louis, MO, USA) at –80°C. Total RNA was extracted using TRIzol™ (Invitrogen, San Diego, CA, USA), according to the manufacturer's protocol. RNA concentration and quality were determined by spectrophotometry (NanoDrop2000, Thermo Fisher Scientific, Madrid, Spain), measuring the absorbance at λ = 260 and 280 nm, and by denaturing electrophoresis in TAE agarose gel (1.5%). For cDNA preparation, total RNA was treated with DNAse I Amplification Grade (Invitrogen, San Diego, CA,

USA) according to the manufacturer's protocol to remove traces of genomic DNA. Total RNA was then reverse transcribed in a 10 µl reaction volume containing 3 µg total RNA using the SuperScript™ First-Strand Synthesis System for RT-PCR (Invitrogen, San Diego, CA, USA) with oligo (dT)$_{12-18}$ (0.5 µg µL$^{-1}$) and random hexamer primers (50 ng µl$^{-1}$), 10X RT buffer [200 mM Tris-HCl (pH 8.4), 500 mM KCL, 25 mM MgCl$_2$, 0.1 M DTT, 10 mM dNTP mix, SuperScript™ II RT (50 U µL$^{-1}$), RNaseOUT™ (40 U µL$^{-1}$)], followed by RNase H (2 U µL$^{-1}$) treatment (Invitrogen, San Diego, CA, USA). Reverse transcription reactions were performed in a thermocycler (Mastercycle R nexus GSX1, Eppendorf AG, Hamburg, Germany) and run according to the manufacturer's protocol. Samples were diluted 1:20 in molecular biology-grade water and stored at –20 °C until further analyses [34]. To obtain *Arapaima* sp.-specific sequences of *amy*, *try*, *ctr*, *pga*, *lpl*, *plA2*, *pyy*, and *elongation factor 1* (*ef1*), alignments of teleost homologues for these gene sequences obtained from GenBank were made using BioEdit Sequence Alignment Editor ver. 7.0.5.2 [35]. Consensus primers designed from the conserved regions identified in these alignments were used for the amplification of *Arapaima* sp. gene sequences. The amplified fragments were separated using 2% agarose gel electrophoresis, and the resulting bands of the expected size were excised, isolated, purified (QIAQuick PCR purification kit, Qiagen, Hilden, Germany), and sequenced [34]. The identity of each sequence was verified using the NCBI BLAST analysis tool (www.ncbi.nlm.nih.gov/BLAST), and the sequences were deposited in GenBank (Table 2).

## Gene expression analyses among digestive tissues and during development

Specific primer sequences for qPCR analyses were designed to amplify products ranging from 80 bp to 200 bp (Table 2). Quantitative PCR analyses for each gene were performed in triplicate using a 7300 Real-Time PCR System (Applied Biosystems, Roche, Barcelona, Spain). The amplification mix consisted of 1 µl cDNA, 0.5 µl primers (20 µM), 10 µl SYBR Green Supermix (Life Technologies, Carlsbad, CA, USA), and molecular biology-grade water, in a total volume of 20 µl. Each set of reactions in every 96-well plate included a 'reagent-only' negative control [34]. The amplification conditions

**Table 2. Accession numbers and oligonucleotide primers used for qPCR analysis of the genes encoding six digestive enzymes and the gut hormone peptide YY in different digestive tissues and during the development of *Arapaima* sp. *Ef1* was used as the reference gene. The amplification efficiency for each gene was approximately 100%.**

| Gene name | Genbank accession number | Primer | Nucleotide sequence (5' – 3') | Tm (°C) | Product size (bp) |
|---|---|---|---|---|---|
| *amy* | OP556572 | qpAMYAgF | TGTGGACAACCACGACAATCAGAG | 70 | 152 |
| | | qpAMYAgR | CTATTCCAACGAAAGCTGGACATG | 70 | |
| *try* | MT006343 | qpTYPAgF | AGCTGCAGTGCCTGCAGATCCC | 72 | 100 |
| | | qpTYPAgR | TCCCTCCAGGTATCCGGCGCAG | 74 | |
| *ctr* | MT006344 | qpCHYAgF | CCTTCAGGACTACACCGGTTTCC | 72 | 170 |
| | | qpCHYAgR | TGAAGACTTTGCCAATAGTCATGG | 68 | |
| *plA2* | MT006346 | qpPHLA2AgF | CAACATGATGCATGCTGGCCAATC | 72 | 100 |
| | | qpPHLA2AgR | CTCTTGCAGGTGATTGTCTTGGTG | 72 | |
| *lpl* | MT006345 | qpLIPAgF | CGAGCCCACTGCCAACGTCATAG | 70 | 135 |
| | | qpLIPAgR | TCGAGTTCCATCATAAGCCAGTTG | 74 | |
| *pga* | MT006342 | qpPEPAgF | CCTGGACAGTCCTTCAATGTCATC | 72 | 200 |
| | | qpPEPAgR | CAGTGTCATATCCCAGAATTCCAG | 70 | |
| *pyy* | MT006347 | qpPPYYAgF | ACCCCGGAGAGGACGCGCC | 68 | 80 |
| | | qpPPYYAgR | TCGTGATGAGGTTGATGTAGTGTC | 70 | |
| *ef1* | QPH36768 | qpEFAgF | GGGAGAGTTCGAGGCTGGTATC | 70 | 115 |
| | | qpEFAgR | GTGGAGCCATCTTGTTGACGCC | 70 | |

*amy*, α-amylase; *try*, trypsinogen; *ctr*, chymotrypsinogen; *plA2*, phospholipase A2; *lpl*, lipoprotein lipase; *pga*, pepsinogen; *pyy*, peptide YY; *ef1*, elongation factor 1.

were as follows: 95 °C for 10 min, 40 cycles of 95 °C for 15 s, and 65 °C for 1 min. A standard curve for determining amplification efficiency (E) was generated by amplifying a series of cDNA dilutions. Real-time PCR efficiencies were determined for each gene from the slopes obtained using Applied Biosystems software by applying the equation $E = 10[-1/\text{slope}]$, where E represents PCR efficiency. The relative gene expression ratio (R) for each gene was calculated according to Pfaffl's formula [36]: $R = (E_{\text{target gene}})^{\Delta Cq \text{ target gene (mean sample - mean reference sample)}} / (E_{\text{reference gene}})^{\Delta Cq \text{ reference gene (mean sample - mean reference sample)}}$, where ΔCq is the difference between the target and reference samples. The reference samples were the stomach and D1 (3.19 ± 0.03 cm TL) for the tissue localisation and developmental analyses, respectively.

Relative gene expression was normalised using *ef1* as a reference, which exhibited no significant variation in expression among samples.

## Statistical analyses

Results were expressed as mean ± standard deviation (S.D.). All data were checked for normality (Kolmogorov–Smirnov test) and homogeneity of variance (Bartlett's test). One-way ANOVA was performed to analyse differences in growth, measurements and counts of various histological structures, and gene expression across different digestive regions and developmental stages [30]. Pairwise comparisons were conducted using the Holm–Sidak method to identify significant differences at $P < 0.05$. Statistical analyses were conducted using SigmaStat 3.0 (Systat Software Inc., Richmond, VA, USA).

## Results

### Growth and survival

The TL and WW results are presented in Fig 1. *Arapaima* sp. specimens exhibited exponential growth over the culture period, following the equations $TL = 3.1119e^{0.0223D}$ ($r = 0.95$) and $WW = 0.1215e^{0.0719D}$ ($r = 0.94$), where D represents the day of culture. Growth in terms of TL remained constant from D1 to D5, then significantly increased from D5 to D7, D9 to D11, and D14 to D17 ($P < 0.05$). Growth in terms of WW was constant from D1 to D5, after which it significantly increased until D17 ($P < 0.05$). The survival rate at the end of the trial was 97 ± 2%.

### Histology of the digestive system

At 2.03 ± 0.07 cm TL, approximately 5–7 days post-hatching, the mouth and anus of *Arapaima* sp. were open, and the digestive system was fully developed, both morphologically and histologically. Fig 2 illustrates the external morphology of the digestive system at 3.19 ± 0.03 cm TL, featuring a 6-mm buccopharyngeal cavity, a 9-mm oesophagus, a 15-mm stomach with gastric glands in the cardiac region, and a long, folded intestinal tract (62 mm) fitting within the abdominal cavity.

**Buccopharyngeal cavity.** The buccopharyngeal cavity at 2.03 ± 0.07 cm TL was lined with a simple flat epithelium and exhibited a single row of conical teeth protruding into the pharyngeal lumen (Fig 3). Round goblet cells, numbering 7–10 per 100 μm of epithelium (8.3 ± 1.5 cells per 100 μm), with a diameter of 6–12 μm (9.9 ± 1.6 μm), covered the epithelium. Setiform taste buds were scattered along the epithelium up to the oesophageal region (Figs 3a, b, and 4a), including the lips and the bases of the gill arches. The number of goblet cells increased towards the oesophagus, while taste buds were particularly abundant in the lips (mainly on the dentary) and distributed uniformly along the buccopharynx (Fig 4b). Both goblet cells and taste buds stained positively with AB and PAS, indicating a richness in sulphated and carboxylated acid mucopolysaccharides and sialomucins (glycoproteins). These histochemical properties remained constant throughout the study. Sensory cells (superficial neuromasts) were also observed on the external surface of the snout, near the nares. The olfactory organ, composed of a simple ciliated epithelium, contained mucous cells that stained positively with AB and PAS (Fig 4c, d).

**Oesophagus.** At 2.03 ± 0.07 cm TL, the oesophagus was characterised by a multilayered squamous epithelium in the anterior region and a simple columnar epithelium in the hind part, lined by a basophilic basal membrane. It featured long

folds densely covered by columnar goblet cells (10–14 cells per 100 μm of epithelium; 12.80±1.26 cells per 100 μm), with a diameter of 8–17 μm (11.83±2.46 μm). These mucous cells contained neutral and acidic (carboxylated and sulphated) mucosubstances (PAS- and AB-positive staining), which were also secreted into the oesophageal lumen (Fig 5a-d). During development, the number of goblet cells increased, concurrently with a decrease in cell diameter (32–34 cells per 100 μm of epithelium; 33.20±0.77 cells per 100 μm; a diameter of 4–14 μm; 9.46±3.19 μm at 5.05±0.34 cm TL). Taste buds were absent in the oesophagus. The oesophageal mucosa was encased by a layer of connective tissue, a circular striated muscle layer, and a thin tunica serosa (Fig 5a-e). The striated muscle layer transitioned to smooth fibres in the circular muscular layer at the junction between the oesophagus and the stomach (Fig 5e). No significant changes were observed in the width of the epithelium (16.9±3.4 μm) and muscular layer (33.7±11.1 μm) throughout the study period. The number and size of mucosal folds increased during development, from 0–1 fold (0.67±0.5) to 2 folds (2±0) per 100 μm of epithelium, and from 99.4±21 μm to 117±26 μm in width, respectively, between 2.03±0.07 and 5.05±0.34 cm TL.

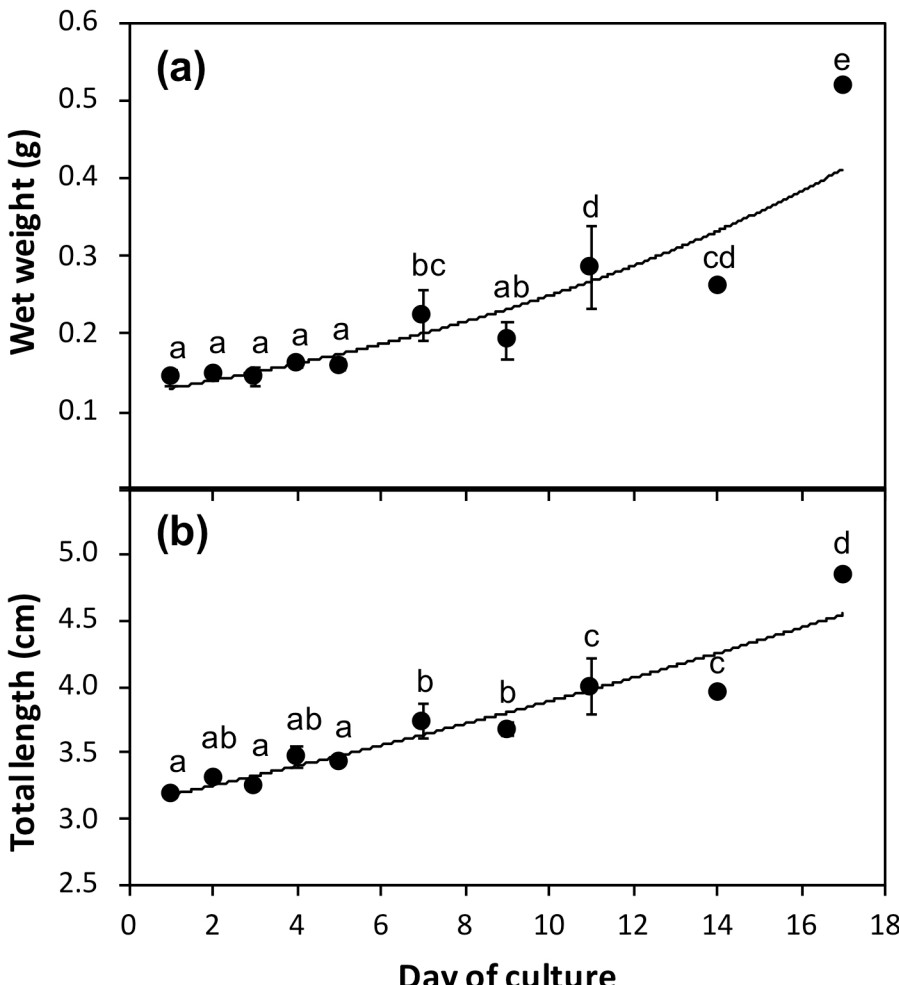

**Fig 1. Growth in terms of total length (TL) and wet weight (WW) of *Arapaima* sp. reared in a clear-water recirculating aquaculture system at 29.0 ±0.03 ºC during the 17-day trial period. Data are expressed as mean±S.D.** Values with different letters indicate significant differences in growth between time points (one-way ANOVA, $P < 0.05$).

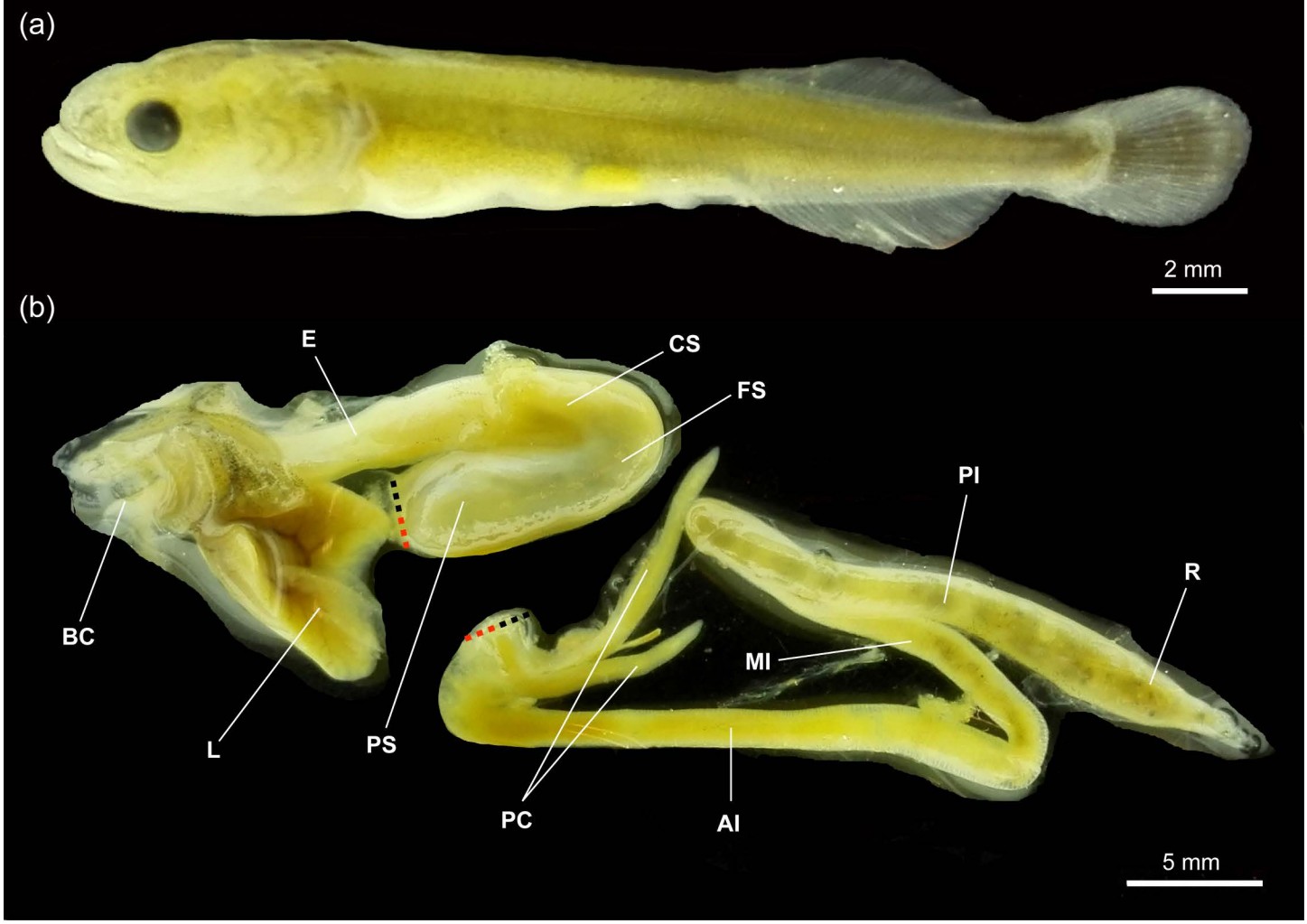

**Fig 2. External morphology (a) and anatomy of the digestive system (b) of an early juvenile (3.19±0.03 cm TL) *Arapaima* sp.** The dotted lines indicate the connection point between the stomach and anterior intestine, which were separated for better illustration. AI, anterior intestine; BC, bucco-pharynx; CS, cardiac stomach; OE, oesophagus; FS, fundic stomach; L, liver; MI, middle intestine; PI, posterior intestine; PC, pyloric caeca; PS, pyloric stomach; R, rectum.

**Stomach.** The J-shaped stomach was fully developed at 2.03±0.07 cm TL and exhibited three distinct gastric regions: cardiac, fundic, and pyloric (Fig 6a,b). The mucosa in these regions consisted of a simple columnar epithelium containing glycogen as well as neutral and acidic glycoconjugates (PAS- and AB-positive) and a connective subepithelial tissue layer (Fig 6c). The epithelial cells had a granular cytoplasm with a basal nucleus and microvilli at the apical border (Fig 6c), and their height varied across the gastric regions. A large number of tubular gastric glands, composed of a distinct type of secretory cell lacking microvilli at the apical border and forming aggregated cells connected to the lumen, were located in the cardiac region (Fig 6b-e). The size (19.7±4.5 µm in diameter) and number (6.3±0.9 cells in 100 µm of mucosa) of these gastric glands remained constant throughout the study period. Mucosubstances were not detected in the gastric glands (Fig 6e). The mucosa of the cardiac region featured longitudinal folds lined by a simple short columnar epithelium (13.3±1.5 µm in height), a thin lamina propria, gastric glands, and submucosa layers. It was encased by a thin circular muscle layer (48.1±4.1 µm in width) and a serosa layer. In the fundic region, the simple, ciliated columnar epithelium and the circular muscular layer were

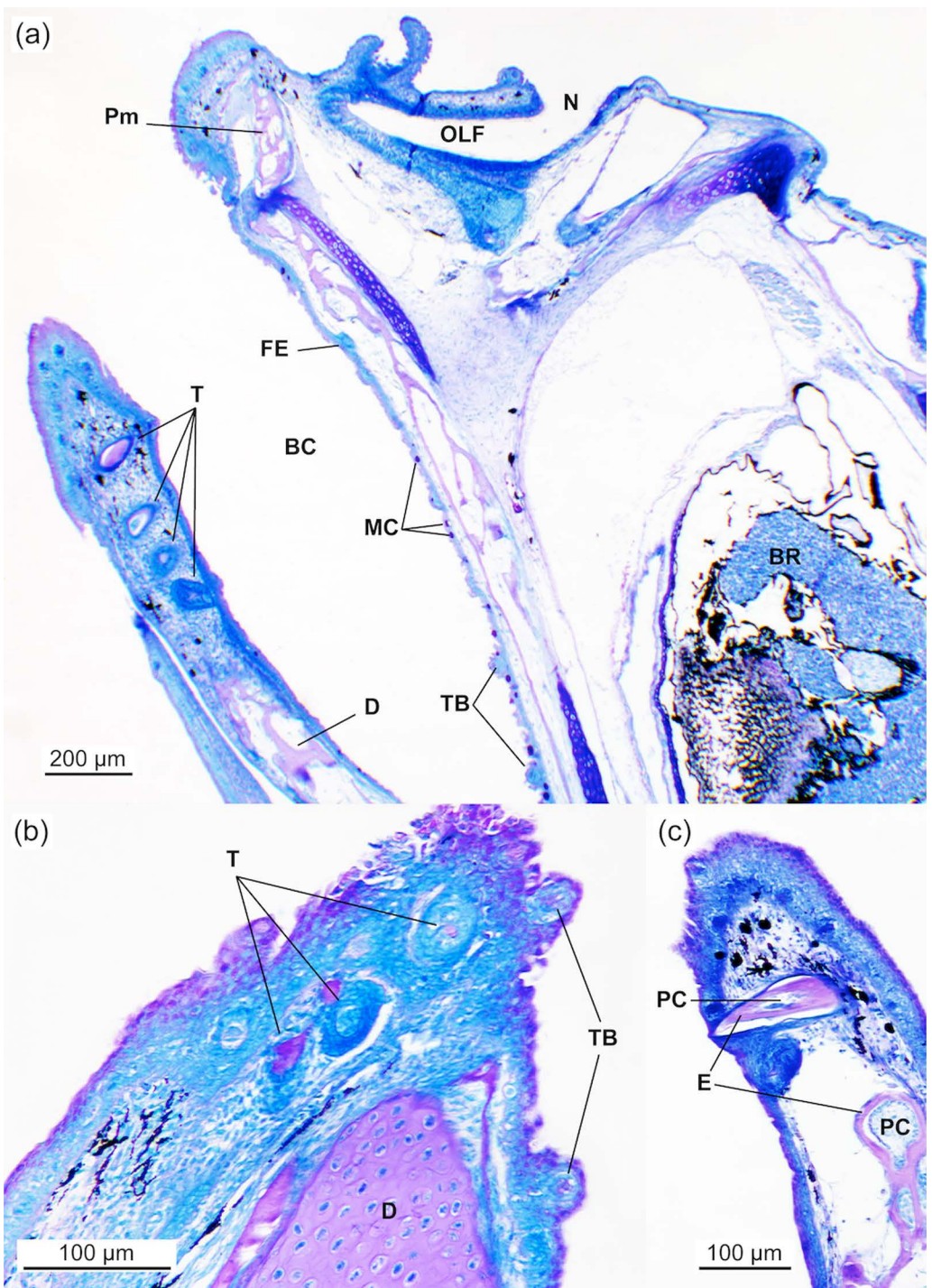

**Fig 3. General view of the buccopharynx and olfactory organ of *Arapaima* sp. (2.03±0.07 cm TL).** (a) General view of the head showing the buccopharynx and the olfactory organ. Note the presence of developing conical teeth in the lower jaw and mucous cells containing neutral mucosubstances (PAS-positive) and taste buds in the buccopharyngeal epithelium. (b) Details of the maxilla and dentary showing developing conical teeth and the presence of setiform taste buds. BC, buccopharyngeal cavity; BR, brain; D, dentary bone; E, enamel; FE, flat epithelium; MC, mucous cell; N, nare; OLF, olfactory organ; PC, pulp cavity; Pm, premaxilla; TB, taste bud; T, tooth. PAS staining.

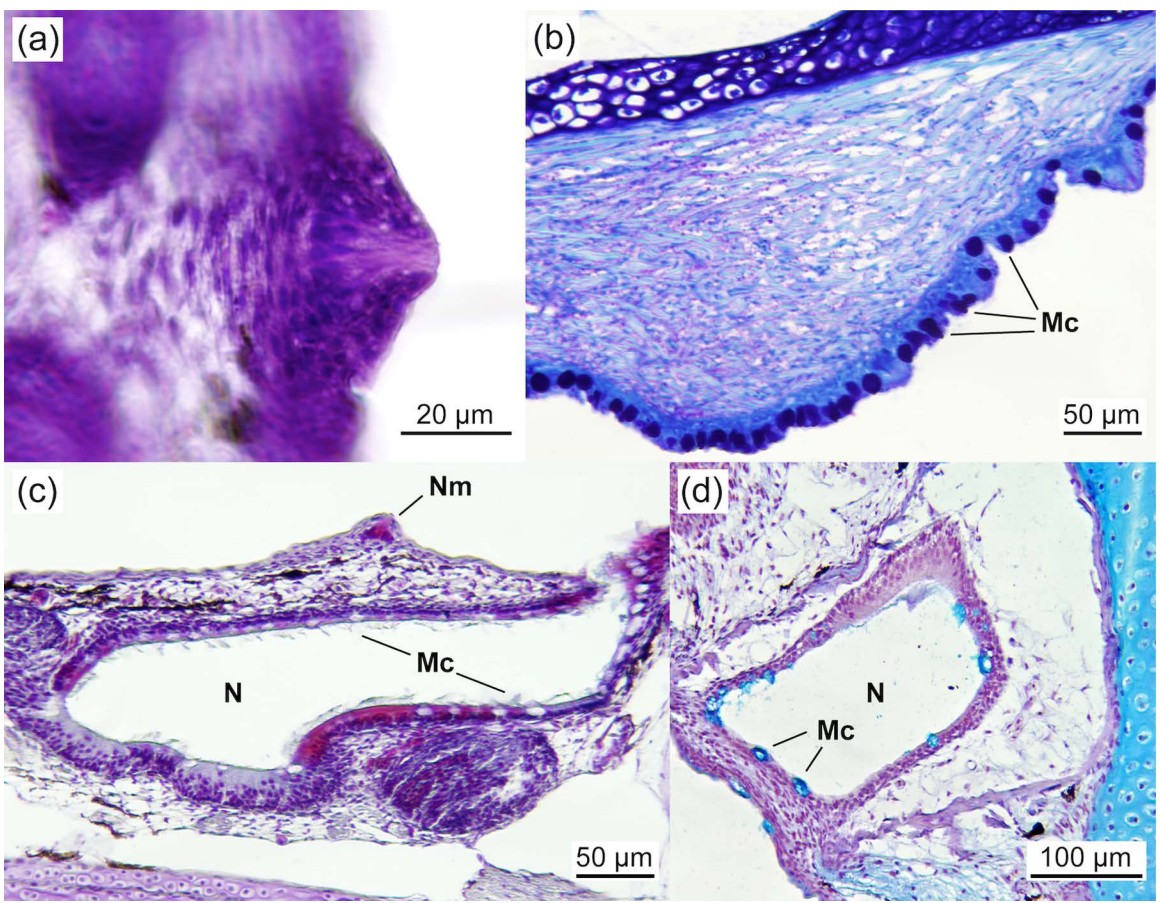

**Fig 4. Detail of the buccopharynx and olfactory organ of *Arapaima* sp .** (a) HE staining showing a setiform taste bud of the buccopharynx (3.43±0.01 cm TL). (b) PAS staining of the epithelium of the buccopharynx near the transition to the oesophagus, covered by a high number of mucous cells containing neutral mucosubstances (PAS-positive) (3.31±0.01 cm TL). (c) VOF staining (3.19±0.03 cm TL) and (d) AB staining (2.03±0.07 cm TL) of the olfactory organ showing superficial neuromasts located on the external surface of the snout and the nare, composed of a simple ciliated epithelium with numerous mucous cells containing both neutral and acidic mucosubstances (AB positive). Mc, mucous cell; N, nare; Nm, neuromast.

taller (25.8±4.9 vs. 13.3±1.5 µm in height) and thicker (204.8±25.7 µm vs. 48.1±4.1 in width), respectively, compared to the cardiac region (Fig 6f). The pyloric region exhibited shorter longitudinal folds than the fundic region, along with a shorter ciliated columnar epithelium (9.1±1.0 µm in height) and a thinner circular muscle layer (53.1±5.1 µm in width), similar to the cardiac region. The pyloric region was connected to the anterior intestine by a well-developed pyloric sphincter (Fig 7a).

   **Intestine.** The intestine of *Arapaima* sp. was long and folded to fit within the abdominal cavity (Fig 2). At 2.03±0.07 cm TL, it was fully developed, consisting of a simple columnar epithelium that was PAS-positive in the brush border. Three distinct intestinal regions were identified: anterior, middle, and posterior (Figs 7b, d, 8, and 9a,b). The anterior intestine had long folds that shortened towards the posterior intestine (Fig 7b, Table 3; *P*<0.05). The number of folds also decreased from the anterior to the posterior intestine, where it resembled undulations (Table 3; *P*<0.05). A significant increase in the number of folds during development was observed only in the middle intestine (Table 3; *P*<0.05). Two main pyloric caeca, measuring 7.3 and 12 mm in a 3.19±0.03 cm TL specimen (Fig 2), branched from the anterior intestine, just posterior to the pylorus. They were smaller in diameter than the intestine (by 1.5 times, 209.5±13.4 µm) and histologically similar to the anterior intestine (Fig 7c).

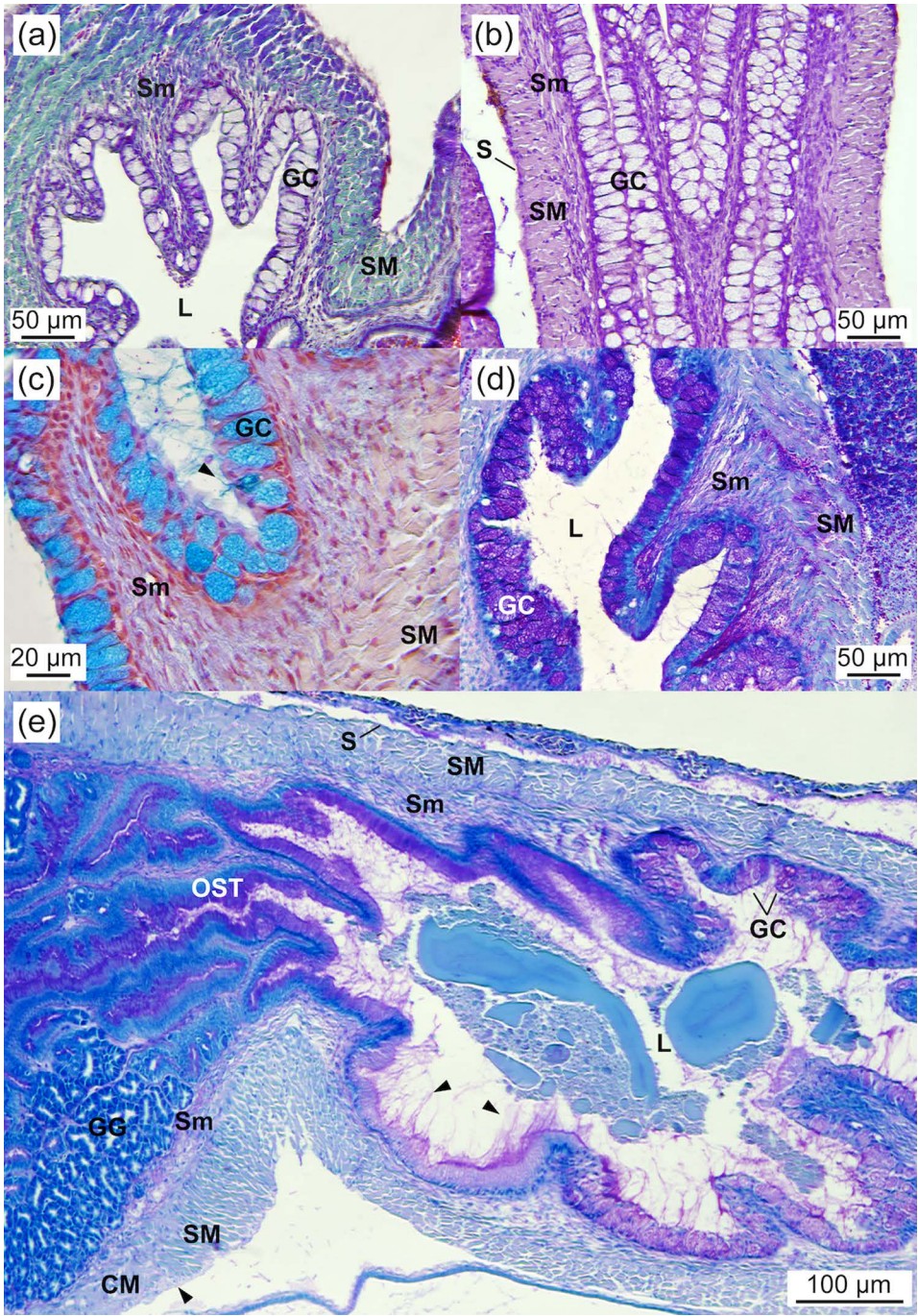

**Fig 5. Oesophagus of *Arapaima* sp.** (a) VOF staining showing a thick layer of striated muscle and a columnar epithelium with numerous columnar mucous cells (3.26 ± 0.02 cm TL). (b) Longitudinal section (HE staining) depicting the different layers of the oesophagus and the long mucosal folds (2.03 ± 0.07 cm TL). (c) Detail of the mucous cells containing and secreting neutral and acidic mucosubstances into the oesophageal lumen (arrowheads) (AB staining) (3.26 ± 0.02 cm TL). (d) Detail of the mucous cells containing and secreting neutral mucosubstances into the oesophageal lumen (PAS staining) (3.26 ± 0.02 cm TL) (e) View of the transition from the oesophagus to the stomach characterised by long, abundant, and intricate mucosal folds (PAS staining) (3.74 ± 0.12 cm TL). Note the large amount of mucosubstances released by mucous cells into the oesophageal lumen and the loss of the striated muscular layer at the oesophagus–stomach transition (arrowheads). CM, circular smooth muscular layer; GC, goblet cell; GG, gastric gland; L, lumen; S, serosa; Sm, smooth muscle; SM, striated muscle; OST, oesophagus–stomach transition.

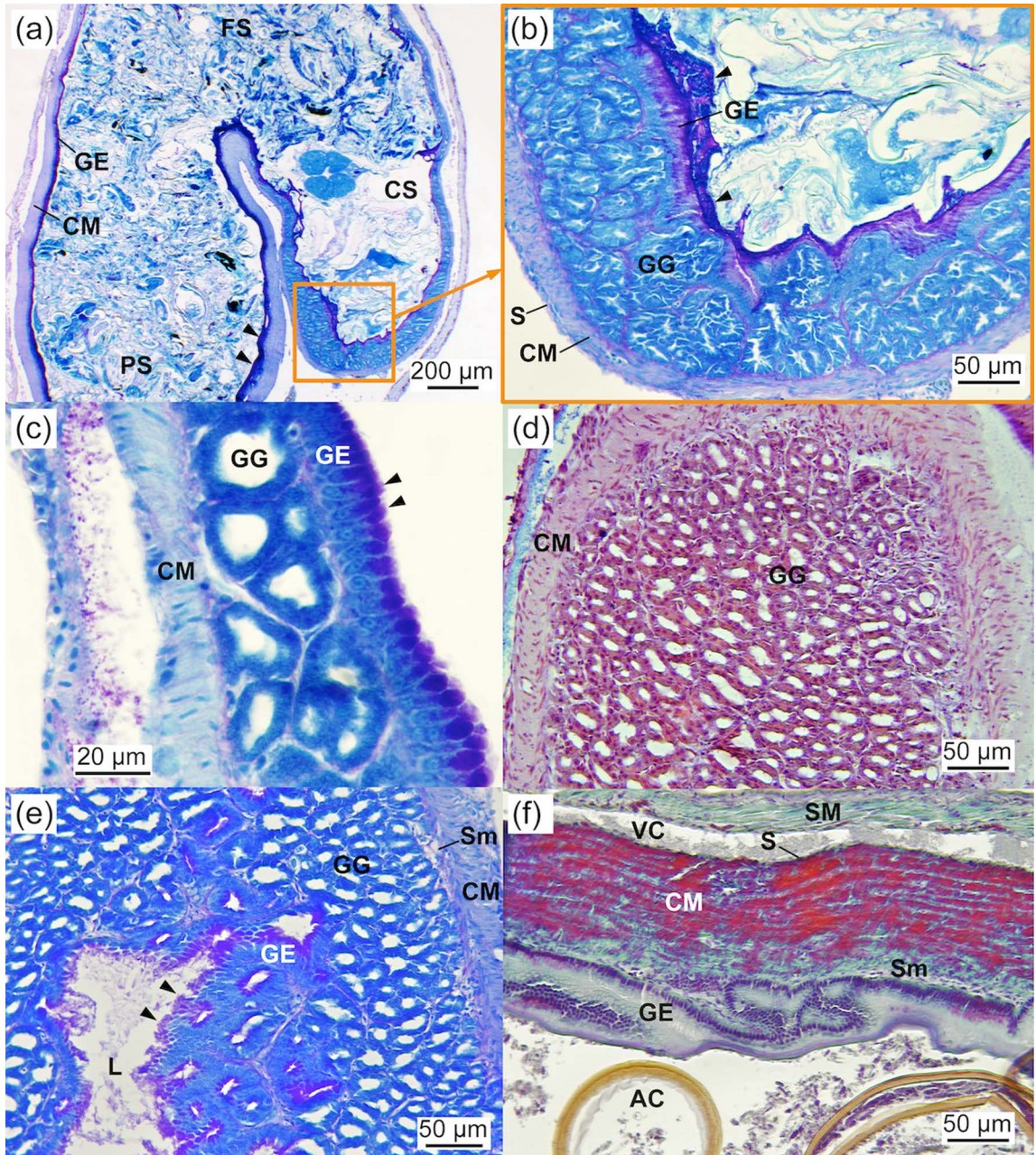

**Fig 6. Stomach of *Arapaima* sp.** (a) General view of the J-shaped stomach showing the cardiac, fundic, and pyloric regions (PAS staining; 2.03±0.07 cm TL). Arrowheads indicate the presence of neutral mucosubstances in the gastric epithelium of the fundic stomach (PAS-positive). (b) Detail of the gastric stomach containing gastric glands at 2.03±0.07 cm TL (PAS staining). Note the secretion of neutral mucosubstances into the lumen of the stomach by the gastric epithelium (arrowheads). (c) PAS staining showing the stomach wall, composed of a simple columnar epithelial layer containing neutral mucosubstances (arrowheads; PAS-positive), a connective subepithelial layer, tubular gastric glands connected to the lumen, and a circular layer of muscle surrounding the gastric mucosa (3.43±0.01 cm TL). (d) AB staining showing the gastric stomach full of gastric glands without mucosubstances (AB-negative; 3.26±0.02 cm TL). (e) PAS staining showing the gastric stomach full of gastric glands showing no signs of neutral mucosubstances (PAS-negative) and the gastric epithelium containing neutral mucosubstances (arrowheads; PAS-positive; 3.26±0.02 cm TL). (f) VOF staining of the fundic region of the stomach, characterised by a thick layer of striated muscle and a tall columnar gastric epithelium with longitudinal folds (3.19±0.03 cm TL). AC, *Artemia* cyst; CM, circular muscle; CS, cardiac stomach; FS, fundic stomach; GE, gastric epithelium; GG, gastric glands; L, lumen; S, serosa; Sm, smooth muscle; SM, striated muscle; VC, visceral cavity.

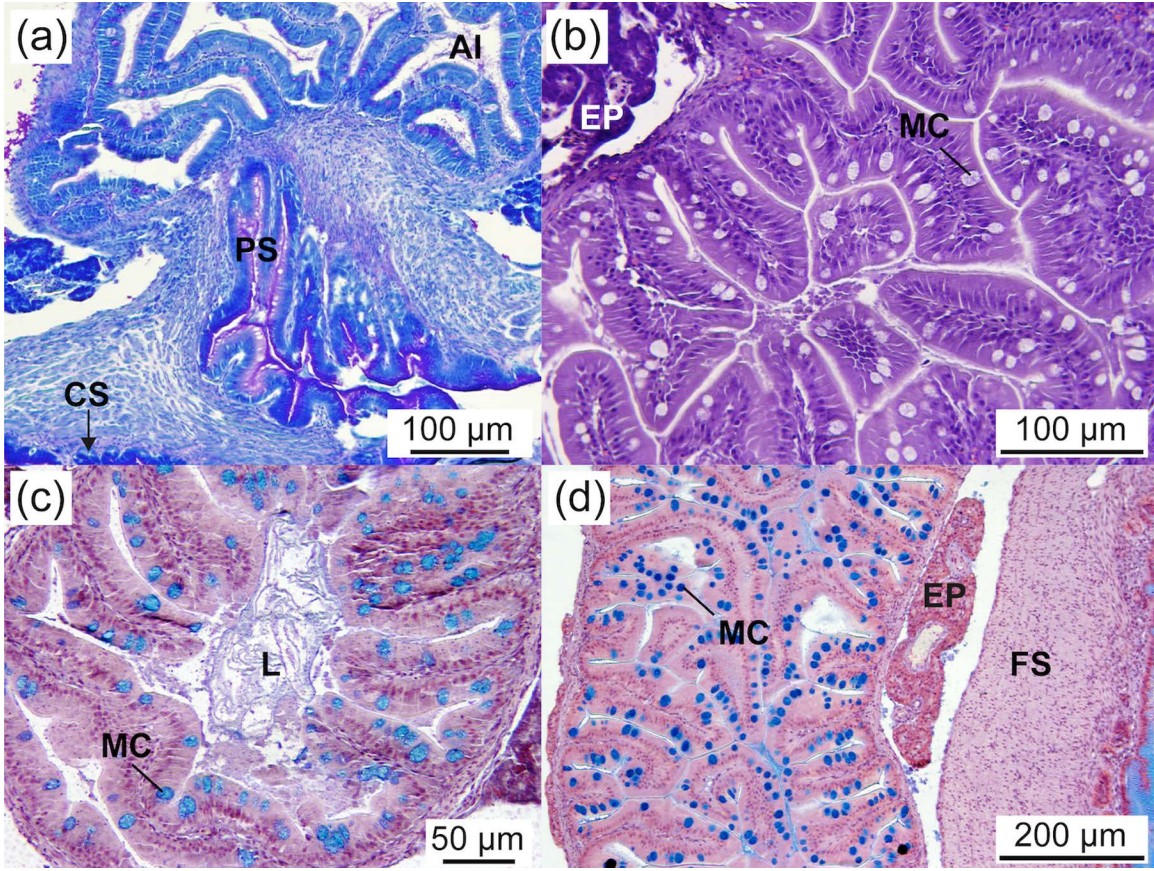

**Fig 7. Anterior intestine of *Arapaima* sp.** (a) Detail of the transition from the stomach to the anterior intestine showing the pyloric sphincter (PAS staining; 3.26±0.02 cm TL). (b) Detail of the mucosa showing numerous folds and mucous cells (HE staining; 2.43±0.07 cm TL). (c) Detail of the mucosa of a pyloric caecum showing numerous mucous cells containing neutral and acidic mucosubstances (AB staining; 2.43±0.07 cm TL). (d) General view of the anterior intestine showing the mucosal folds and epithelial mucous cells containing neutral and acidic mucosubstances. Note the exocrine pancreas surrounding the anterior intestine (AB staining; 4.00±0.2 cm TL). AI, anterior intestine; CS, cardiac stomach; EP, exocrine pancreas; FS, fundic stomach; L, lumen; MC, mucous cell.

The intestinal wall comprised the serosa, muscular (smooth circular), submucosa, and mucosa layers. The intestinal epithelium featured a monostratified columnar epithelium with nuclei located mid to basal and a microvilli layer (brush border) at the apical surface. Throughout the study, the number of enterocytes was higher in the posterior intestine than in the anterior and middle regions (Table 3; $P<0.05$; Figs 7b and 8b). The number of enterocytes in the middle intestine increased with age, along with the number of folds (Table 3; $P<0.05$). The enterocyte size (height) was greater in the anterior intestine than in the middle and posterior regions at 2.03±0.07 cm TL, but this difference disappeared by the end of the study due to a subsequent decrease in enterocyte size in the anterior intestine and an increase in the posterior intestine (Table 3; $P<0.05$). At 2.03±0.07 cm TL, the decrease in enterocyte size from the anterior to the posterior intestine corresponded with an increase in cell number, whereas at 5.05±0.34 cm TL, the enterocyte size remained consistent regardless of cell number (Table 3). There was a positive correlation between the number of folds and size increase across the intestinal regions ($r=0.99$, $P=0.01$). Intracellular lipid inclusions at 2.03±0.07 cm TL were most abundant in the middle intestine, followed by the anterior and posterior regions (Table 3; $P<0.05$). However, at 5.05±0.34 cm TL, the anterior intestine had a higher number of lipid inclusions than the posterior intestine, with intermediate values in the

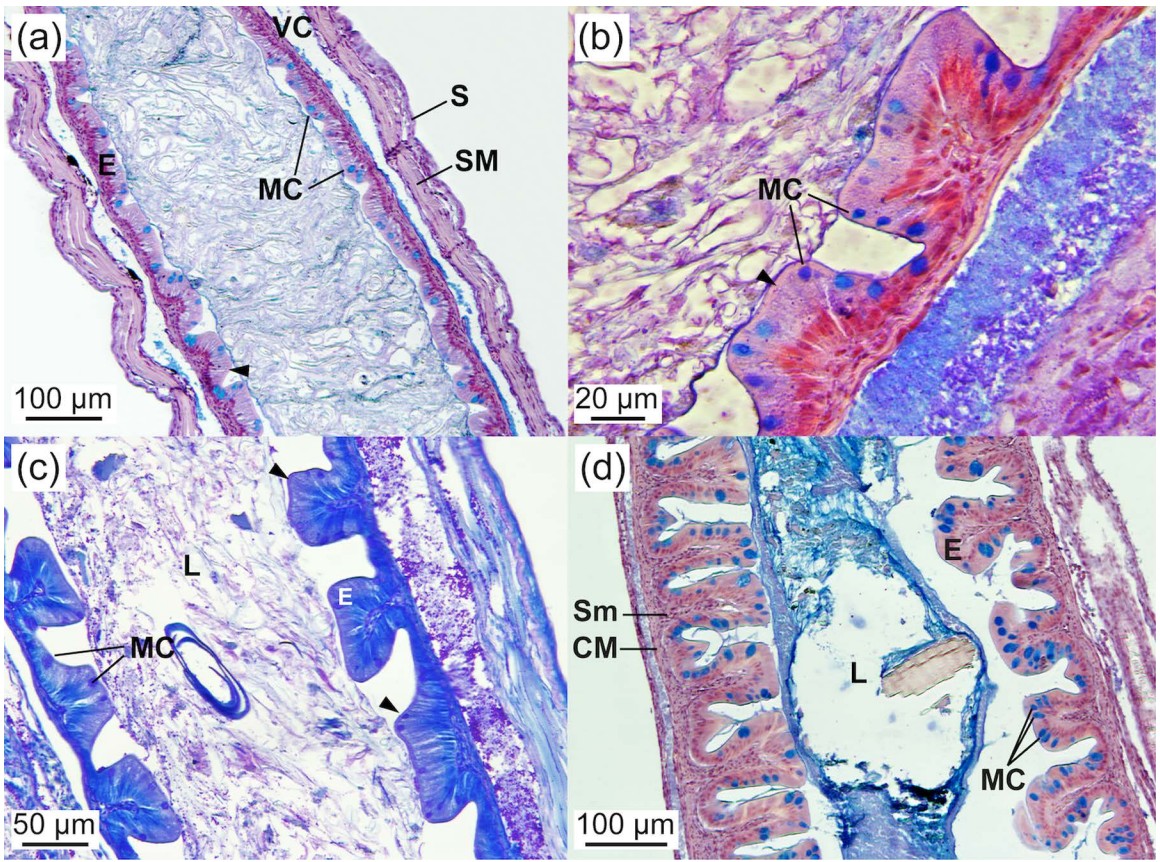

**Fig 8. Middle intestine of *Arapaima* sp.** (a) General view of the middle intestine, showing mucosa with shorter folds than in the anterior intestine and mucous cells containing neutral and acidic mucosubstances (AB staining; 4.00±0.2 cm TL). (b) Detail of the epithelial mucosa, with functional mucous cells in direct contact with food (arrowhead) (AB staining; 3.26±0.02 cm TL). (c) PAS staining showing epithelial mucous cells containing neutral mucosubstances secreted into the lumen (arrowheads) (PAS-positive; 3.26±0.02 cm TL). (d) General view of the middle intestine wall, showing its different layers (AB staining; 4.00±0.2 cm TL). CM, circular muscle; E, epithelium; L, lumen; MC, mucous cell; S, serosa; Sm, smooth muscle; SM, striated muscle.

middle intestine (Table 3; $P < 0.05$). The number of lipid deposits in the middle and posterior intestines decreased with age (Table 3; $P < 0.05$). The surface area of lipid deposits at 2.03±0.07 cm TL was similar across all three intestinal regions (averaging 16.7 µm²), but it significantly increased from the anterior to the posterior region at 5.05±0.34 cm TL (Table 3; $P < 0.05$). The surface area of the lipid droplets decreased with age in the anterior intestine and increased in the posterior intestine (Table 3; $P < 0.05$). The size of the lipid deposits in the middle intestine varied greatly among individuals during development, resulting in no significant age-related differences (Table 3; $P > 0.05$). Goblet cells developed progressively throughout the experimental period and were present in all intestinal regions, including the pyloric caeca, and were most abundant in the posterior intestine, followed by the middle and anterior intestines (Table 3; $P < 0.05$; Figs 7b-d, 8, and 9b). The number of goblet cells in the middle intestine increased with age, concurrent with an increase in the number of enterocytes, intestinal folds, and lipid droplets (Table 3; $P < 0.05$). A positive correlation was observed between the increase in enterocyte and goblet cell numbers across intestinal regions ($r = 0.97$, $P = 0.02$). Mucous cells stained positive for PAS and AB, indicating the presence of neutral and acidic mucosubstances. No differences were detected in the histochemical properties of mucin in the intestinal goblet cells across intestinal regions.

**Liver.** At 2.03±0.07 cm TL, the liver of *Arapaima* sp. was bilobulated, comprising polyhedral hepatocytes with large, centrally located basophilic nuclei and eosinophilic granular cytoplasm containing a high glycogen content (Fig 10). The

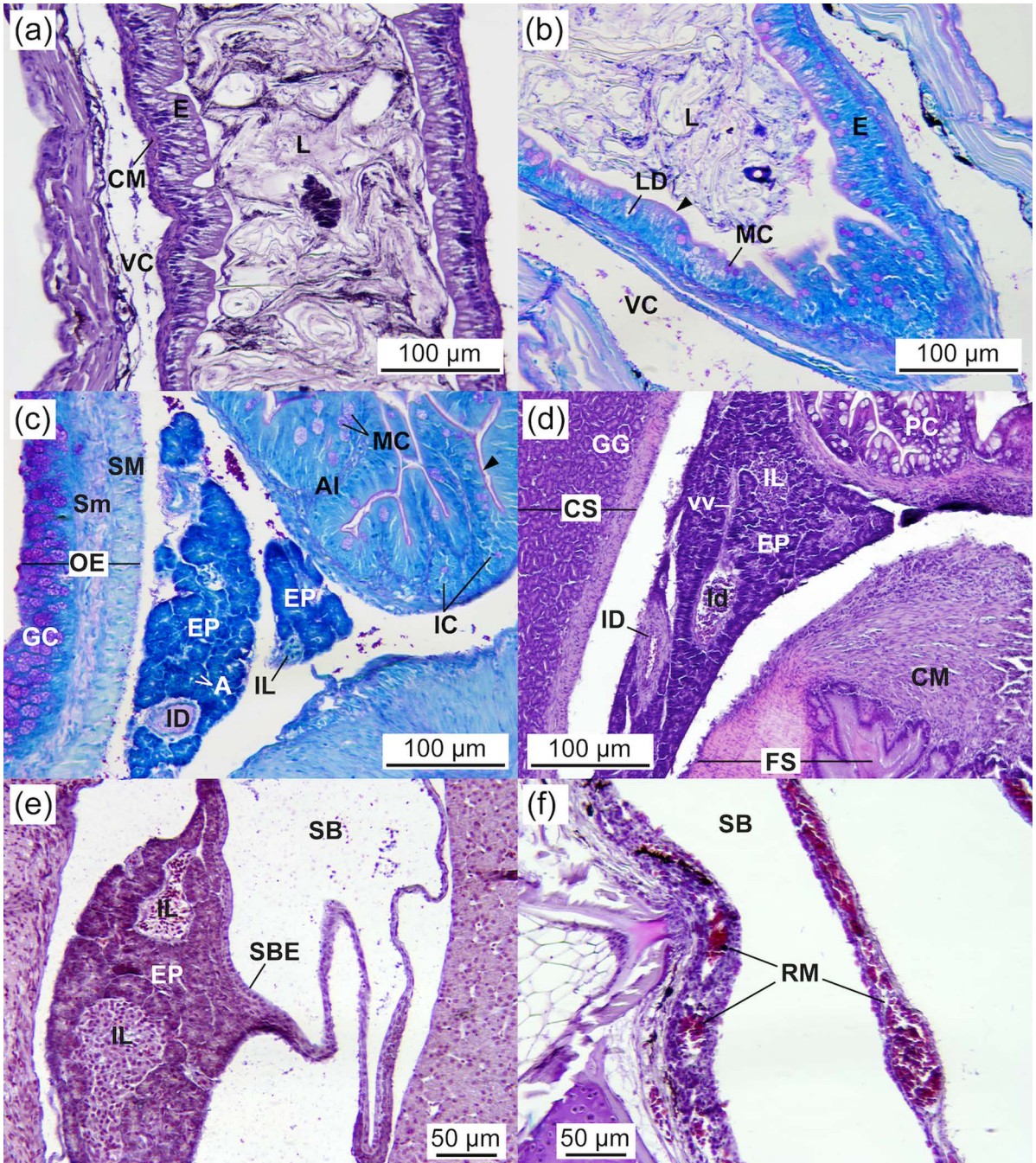

**Fig 9. Posterior intestine, exocrine pancreas, and swim bladder of *Arapaima* sp.** (a) General view of the posterior intestine, showing short folds and a simple columnar epithelium containing mucous cells (HE staining; 2.43±0.07 cm TL). (b) Detail of the rectum, showing an epithelium with mucous cells containing neutral and acidic mucosubstances, which are secreted into the lumen (arrowheads), and the presence of lipid droplets. (c) Exocrine pancreas surrounding the anterior intestine. Note the mucosubstances being secreted into the intestinal lumen (arrowhead) (PAS staining; 2.03±0.07 cm TL). (d) Exocrine pancreas surrounding a pyloric caecum (HE staining; 5.05±0.34 cm TL). (e) Exocrine pancreas surrounding the swim bladder (AB staining; 2.43±0.07 cm TL). A, acini; AI, anterior intestine; CS, cardiac stomach; CM, circular muscle; EP, exocrine pancreas; FS, fundic stomach; GC, goblet cells; GG, gastric glands; IC, intestinal crypt; ID, interlobular duct; Id, intralobular duct; IL, islet of Langerhans; MC, mucous cell; OE, oesophagus; RM, rete mirabile; SB, swim bladder; SBE, swim bladder epithelium; Sm, smooth muscle; SM, striated muscle; vv, vein vessel.

Table 3. Number and size of different components of the intestinal mucosa of *Arapaima* sp. specimens at 2.03±0.07 and 5.05±0.34 cm TL, reared at 29.0±0.03 ℃. Data are expressed as the mean±S.D. Different letters within columns denote statistically significant differences among intestinal sections within the same developmental stage (one-way ANOVA, *P*<0.05), and asterisks denote statistically significant differences between developmental stages within the same intestinal section (t-test, *P*<0.05).

| | Number of enterocytes (in 100 μm²) | Height of enterocytes (μm) | Number of folds (in 100 μm²) | Length of folds (μm) | Number of lipid deposits (in 100 μm²) | Surface of lipid deposits (μm²) | Number of goblet cells (in 100 μm²) |
|---|---|---|---|---|---|---|---|
| *Anterior intestine* | | * | | | | * | |
| 2 cm TL | 19.5±1.7b | 32.2±3.9a | 1.6±0.5a | 123.2±36.9a | 18.1±2.7b | 18.9±6.2a | 4.5±0.6b |
| 5 cm TL | 21.7±0.8b | 24.4±2.9a | 2.0±0.0a | 143.8±63.5a | 15.4±4.3a | 1.3±0.5b | 4.8±0.9b |
| *Middle intestine* | * | | * | | * | | * |
| 2 cm TL | 20.4±1.8b | 17.8±2.1b | 1.0±0.0b | 54.6±18.2b | 43±7.1a | 20.8±81a | 4.3±1.2b |
| 5 cm TL | 26.6±2.4b | 27.2±3.8a | 2.0±0.0a | 77.7±6.5b | 4.5±37ab | 9.7±9.6b | 7.8±1.0b |
| *Posterior intestine* | | * | | | * | * | |
| 2 cm TL | 31.5±0.7a | 13.7±2.6b | 0.0±0.0c | 0.0±0.0c | 8.0±2.0c | 10.5±4.3a | 11.7±1.3a |
| 5 cm TL | 34±2.5a | 24.2±3.0a | 0.0±0.0b | 0.0±0.0c | 2.5±0.8b | 27.7±4.1a | 11.3±1.2a |

level of fat deposits varied among individuals and across ages, ranging from 0 to 100 (45.9±36.5) vacuoles per 100 μm². The size of lipid vacuoles remained constant throughout the study period (13.4±6.2 μm²; *P*>0.05). The degree of lipid inclusion influenced the position of the nuclei within the cells; the higher the lipid content, the more peripheral the nuclei. The number of hepatocytes, organised in cords between sinusoids, often around veins, increased with age from 90 (90.5±8.3) to 169 (169.4±13.5) cells per 100 μm². A negative correlation was observed between the number of hepatocytes and the diameter of lipid deposits (*r*=-0.99, *P*=0.015).

**Pancreas.** The pancreas of *Arapaima* sp. was diffuse and distributed around the stomach, pyloric caeca, and anterior intestine (Fig 9c-e). Fully differentiated and functional at 2.03±0.07 cm TL, the exocrine pancreas consisted of pancreatic cells similar in shape to hepatocytes. These cells were arranged in acini, grouped in rosette patterns, and contained acidophilic (PAS-positive) zymogen granules (Fig 9c). The number of pancreatic cells increased with age from 120 to 390 cells per 100 μm (*P*<0.05).

**Swim bladder.** The swim bladder, located ventrally to the vertebral column, occupied the anteroposterior part of the mesentery. Its epithelium was composed of simple cuboidal cells, surrounded by fibroblasts, and featured a well-developed rete mirabile (Fig 9e,f).

## Gene expression

**Localisation and quantification of gene expression across digestive tissues.** The expression of all analysed genes was detected in all digestive tissues, albeit at variable levels. The expression of *amy, try*, and *ctr* was higher in the liver and anterior intestine, whereas *pga* was predominantly expressed in the stomach (*P*<0.05; Fig 11). The expression of *lpl* was highest in the anterior intestine, followed by the stomach and liver (*P*<0.05), whereas *plA2* expression was more pronounced in the liver and posterior intestine (*P*<0.05). Finally, *pyy* expression was significantly higher in the middle intestine, followed by the anterior intestine and liver (*P*<0.05).

**Gene expression throughout development.** All genes analysed were detected from D1 of the experiment (3.19±0.03 cm TL), and their relative expression levels at D1 were significantly higher than at D3 (3.31±0.01 cm TL, *P*<0.05) (Fig 12). From D3, the expression levels of all genes stabilised, suggesting a response to dietary and environmental changes following the transfer of juveniles from the pond to the recirculating aquaculture system. The

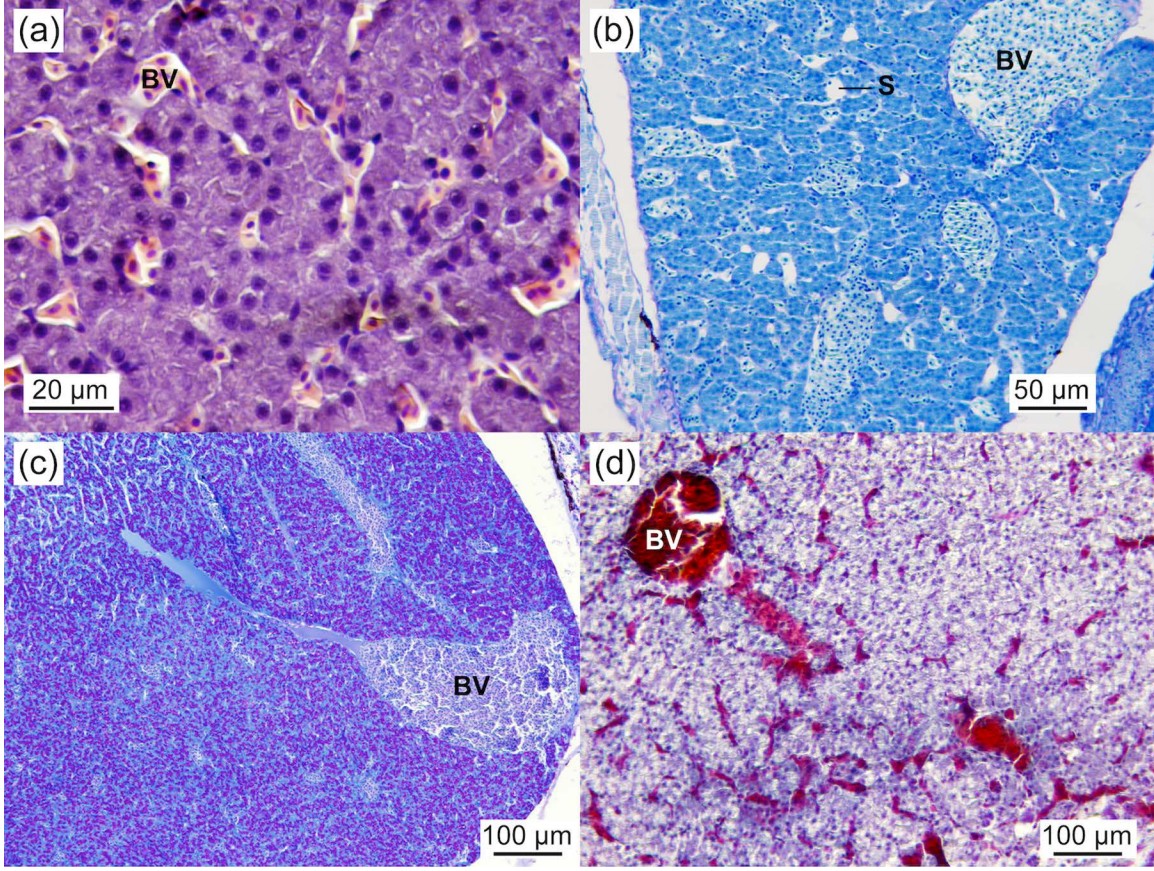

**Fig 10. Liver of *Arapaima* sp. with a prominent vascular system.** Polyhedral hepatocytes are arranged in a cord-like pattern between the sinusoids (a-d). The eosinophilic cytoplasm of hepatocytes (a) indicates macromolecule storage, mainly glycoproteins (c). Hepatocytes have centrally located nuclei and no lipid deposits (a-d). (a) HE staining, 2.03±0.07 cm TL. (b) PAS staining, 3.19±0.03 cm TL. (c) PAS staining, 3.74±0.13 cm TL. (d) VOF staining, 3.19±0.03 cm TL. BV, blood vessel; S, sinusoid.

expression of *amy*, *try*, and *ctr* exhibited a similar pattern during development, with a tendency for lower expression at D5 (3.43±0.01 cm TL) when fish were completely weaned onto the compound diet. However, only *ctr* showed statistically significant differences (4.5-fold decrease; $P < 0.05$). The expression of *pga* increased with age, particularly from D10 to D15 (5.05±0.34 cm TL; 7-fold increase; $P < 0.05$). The expression of *lpl* remained similar between D3 and D5 ($P > 0.05$) and then increased at D10 (4.00±0.2 cm TL, $P < 0.05$), remaining constant thereafter. Both *plA2* and *pyy* expression levels were stable from D3 to D10 ($P > 0.05$) and increased from D10 to D15 (3- and 3.5-fold increase, respectively; $P < 0.05$).

## Discussion

This study demonstrates the successful early weaning of *Arapaima* sp. juveniles onto a compound diet within three days from 3.19±0.03 cm TL, indicating the suitability of the formulated feed (60% crude protein, 15% crude lipids, and 1,653 KJ) for this developmental stage. This rapid adaptation contrasts with previous reports of low acceptance of formulated feeds in this species, which often necessitates gradual feed training [37,38]. The observed growth performance (significant increases in TL and WW) suggests that the compound diet was more effective than *Artemia* nauplii at this developmental stage, potentially due to its optimized nutrient composition, taste, particle size, and compatibility with the digestive capacities of juvenile *Arapaima* sp. Comparable growth and survival were reported in

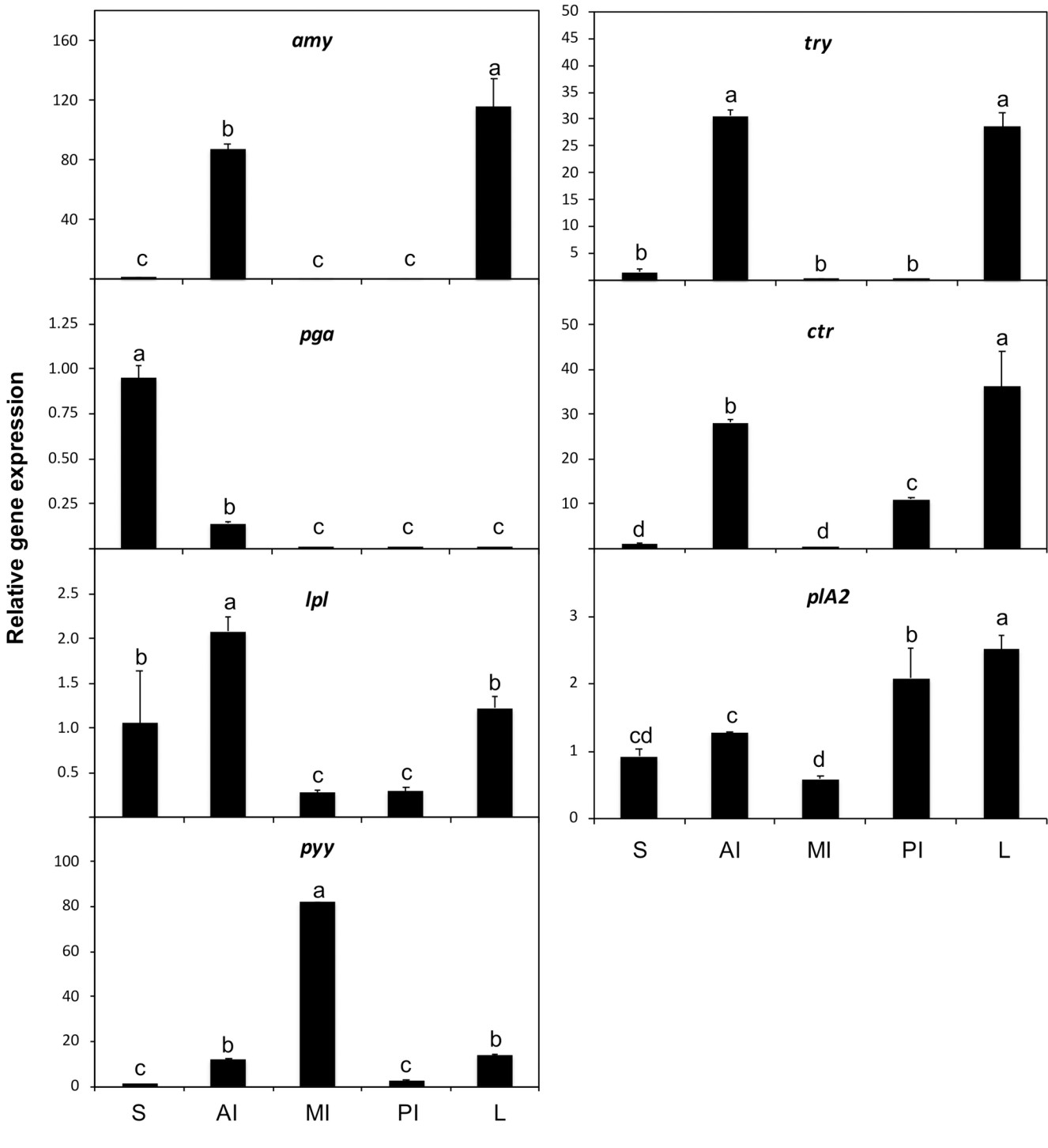

**Fig 11. Relative expression of *α-amylase* (*amy*), *phospholipase A2* (*plA2*), *lipoprotein lipase* (*lpl*), *trypsinogen* (*try*), *chymotrypsin* (*ctr*), *pepsinogen* (*pga*), and *neuropeptide YY* (*pyy*) genes in the stomach (S), anterior intestine (AI), middle intestine (MI), posterior intestine (PI), and liver (L) of 3.19±0.03 cm TL individuals of *Arapaima* sp.** Data are represented as mean±S.D. (*n*=6). Values with different letters denote significant differences in gene expression among digestive tissues (one-way ANOVA, *P*<0.05).

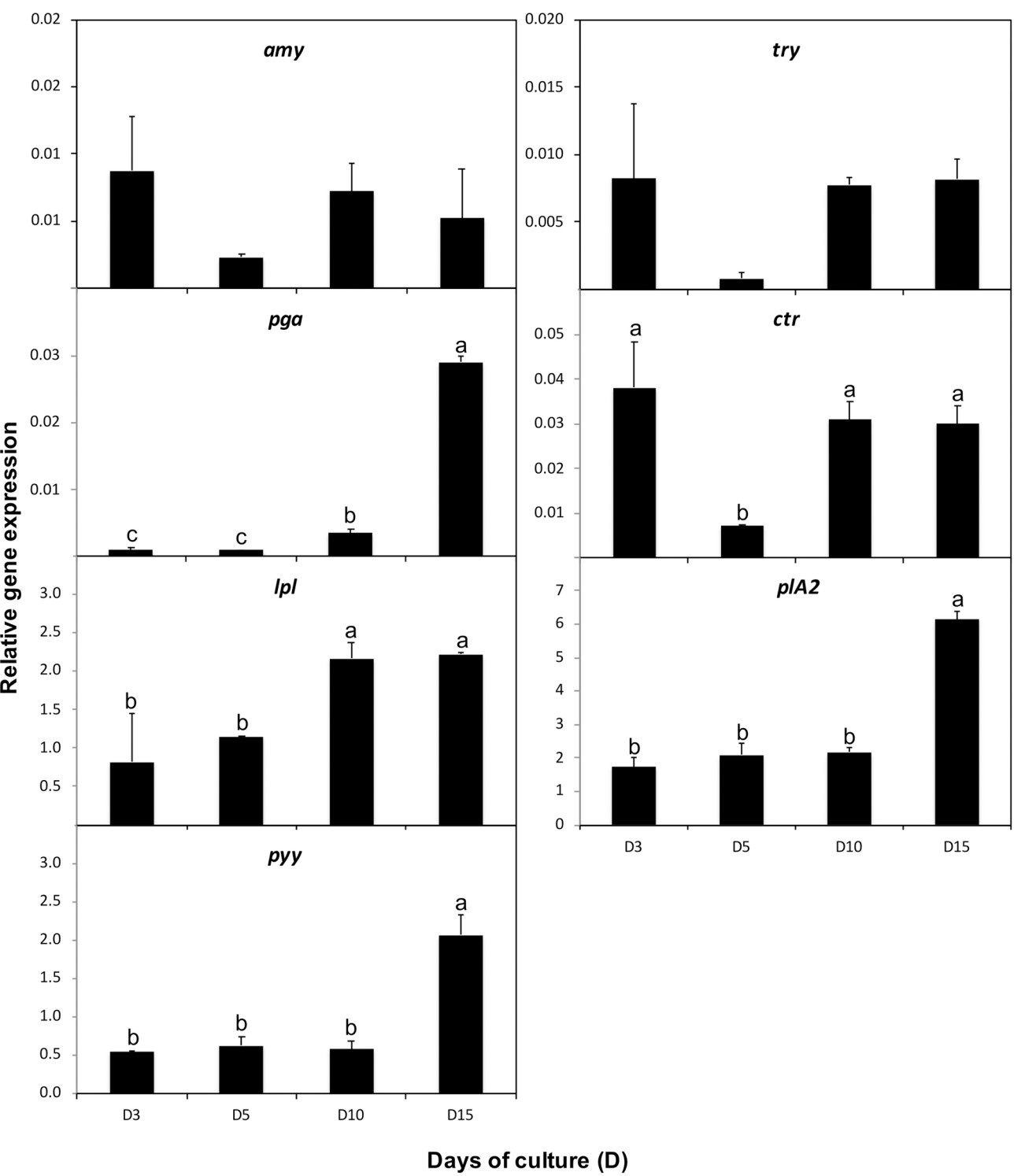

**Fig 12. Relative expression of *α-amylase* (*amy*), *phospholipase A2* (*plA2*), *lipoprotein lipase* (*lpl*), *trypsinogen* (*try*), *chymotrypsin* (*ctr*), *pepsinogen* (*pga*), and *neuropeptide YY* (*pyy*) genes during the development of *Arapaima* sp. reared at 29 °C.** Data are represented as means ± S.D. (*n* = 6). Values with different letters denote significant differences in gene expression during development (one-way ANOVA, *P* < 0.05).

another study using *Artemia* or cladoceran-rich zooplankton fed nine times per day [27], highlighting the efficiency of our compound diet and reduced feeding frequency (five times per day). The high fishmeal content (62%) and the inclusion of protein hydrolysates (17%) and marine phospholipids (7%) likely contributed to these positive outcomes. This formulation can serve as a foundation for further refinement of compound feeds specific to *Arapaima* sp., particularly by leveraging the species' omnivorous tendencies through the inclusion of fishmeal substitutes, with the aim of improving the overall sustainability of its aquaculture.

Our results revealed that the digestive system of *Arapaima* sp. early juveniles is mature upon first surfacing (approximately 2 cm TL at 27 °C), with complete yolk sac resorption, consistent with findings for *Arapaima gigas* in Brazil [39]. The detection of zymogen granules in pancreatic cells at this size indicates the presence of digestive enzyme transcripts. This suggests that the functional ontogeny of the digestive system is synchronised with morphological development, aligning with observations in other fast-growing neotropical fish species [34,40–42]. However, while the histological organization of the oesophagus, stomach, and accessory digestive glands remained consistent after 2.03 ± 0.07 cm TL, increases in cell number and size in the pancreas and intestine, along with increased expression of *lpl*, *plA2*, and *pga*, suggest continued enhancement of digestive and absorptive capabilities up to 5.05 ± 0.34 cm TL.

Morphological and functional characteristics of the digestive system of *Arapaima* sp. are consistent with those of carnivorous species. These characteristics include: (i) a toothed buccopharynx and bony tongue for prey capture [43]; (ii) conical jaw teeth for grasping prey [44]; (iii) long gill rakers typical of piscivorous species to prevent prey escape, in contrast to stubby rakers in omnivorous species [43,44]; (iv) a J-shaped stomach with gastric glands and a short, twice-folded intestine; (v) an increasing density of goblet cells from the anterior to the posterior intestine [45–48]; and (vi) increasing *pga* expression with development for more efficient acidic protein digestion in the stomach [49]. However, unlike typical carnivores, no decrease in *amy* expression was observed [50]. The high *amy* expression, compared to other pancreatic enzymes, suggests a capacity to digest high-carbohydrate diets, consistent with omnivorous feeding habits [34]. The relative gut length (0.42 at 3.19 ± 0.03 cm TL) falls within the ranges of omnivorous species preferring animal prey (0.21–4.3) and carnivorous species with a preference for decapods and fish (0.24–1.64) [51]. These characteristics are consistent with reports of juvenile and adult *Arapaima* spp. exhibiting omnivorous feeding habits but primarily relying on fish [10–13].

Sensory and structural adaptations for bottom feeding, observed in larger *Arapaima* spp. [13], were already present in early juveniles. These included abundant taste buds and mucous cells on the lips and at the bases of the gill arches to detect and trap food particles, respectively, as well as and long gill rakers for filtering [52].

*Arapaima* sp. fingerlings exhibited long, abundant, and intricate mucosa folds, rich in goblet cells secreting substantial amounts of acidic and neutral mucosubstances. These secretions, together with digestive enzymes and hormones, facilitate efficient digestion, aid in the absorption of easily digested substances (such as disaccharides and short-chain fatty acids), support transport processes, protect against mechanical damage and pathogens, contribute to osmoregulation and cell growth, and are essential for survival under controlled aquaculture conditions [53–56]. The observed oesophageal morphology, along with the high survival rates, suggests a potential link to the effectiveness of the feeding protocol.

The location of gastric glands in the cardiac region of the stomach is consistent with several carnivorous species [57–60], while in other species, they are found in the fundic region [61–63], both the cardiac and fundic regions [64], the fundic and pyloric regions [65], or even throughout the stomach [66]. These interspecific differences in gastric gland localisation suggest diverse feeding and digestion strategies [57]. The gastric epithelium of *Arapaima* sp. contained neutral mucosubstances that protect the stomach from autodigestion [67]. Similar neutral glycoconjugates have been identified in the stomach of other fish species [57,58] and are linked to the absorption of easily digestible compounds, such as disaccharides and short-chain fatty acids [45]. This could explain the gene expression of several pancreatic enzymes, particularly *plA2* and *lpl*, in the stomach of *Arapaima* sp., although further research is needed to confirm absorptive processes in the gastric mucosa of this species.

The middle intestine of *Arapaima* sp. appears to be a key site for lipid digestion and absorption, evidenced by the significant *lpl* and *plA2* expression and the observed high number of intracellular lipid deposits. Further research is necessary to fully elucidate the roles of pancreatic and carboxyl ester lipases in the digestion of dietary lipids in this species. The co-expression of *lpl*, *plA2*, and *pyy* in the middle intestine suggests a potential link between gut sensing and gut-brain communication [68], with the high *ppy* expression indicating a role in feed intake regulation through its anorexigenic action [69–73]. In fish, higher *ppy* expression levels have been observed in the stomach, pyloric caeca, anterior intestine, and liver, with lower levels in the posterior intestine [73]. In our study, *ppy* expression was notably higher in the middle intestine, followed by the anterior intestine and the liver. These differences in tissue localisation reinforce the idea that the role of PPY in feeding regulation may be species- and tissue-specific in fish [71].

The decreasing size and number of intestinal folds in the posterior intestine, along with fewer lipid deposits, suggest limited absorption in this region, while abundant mucous cells likely facilitate faecal excretion [74]. In this scenario, the high *plA2* expression in the posterior intestine may be related to intestinal immunity [75,76].

The detected expression of pancreatic enzymes in both the anterior intestine and the liver is consistent with the location of the diffuse pancreas. The inverse gene expression pattern between *ppy* and the pancreatic enzymes may reflect PYY's inhibitory effect on pancreatic secretion [77].

## Conclusions

The histological characterisation and digestive gene expression patterns, along with their tissue localisation, indicate that *Arapaima* sp. early juveniles possess a digestive physiology consistent with that of an omnivorous species with a carnivorous preference. The digestive system is mature from first surfacing, although digestive efficiency continues to improve until at least 5.05±0.34 cm TL. The middle intestine plays a key role in fatty acid absorption and feed intake regulation. Successful weaning onto compound diets was achieved from 3.19±0.03 cm TL. These findings demonstrate the feasibility of a compound diet-based feeding protocol from the point of juvenile harvesting for on-growing, providing valuable insights into the early digestive development and feeding physiology of *Arapaima* sp. in the Peruvian Amazon and contributing to the development of improved feeding and rearing protocols for enhanced *Arapaima* spp. aquaculture. Future research should explore the interplay between optimised compound feed formulations and refined early rearing protocols to maximise growth and survival of this species throughout development.

## Supporting information

**S1 File. Raw growth data of *Arapaima* sp. during the 17-day trial.** Raw total length (cm) and wet weight (g) data of *Arapaima* sp. reared in a clear-water recirculating aquaculture system at 29.0±0.03 ºC over a 17-day period. Data correspond to results presented in Fig 1.
(XLSX)

**S2 File. Raw intestinal histomorphology data of *Arapaima* sp. juveniles.** Raw data on the number and size of different components of the intestinal mucosa of *Arapaima* sp. specimens at 2.03±0.07 and 5.05±0.34 cm TL, reared at 29.0±0.03 ºC. Data correspond to results presented in Table 3.
(XLSX)

**S3 File. Raw gene expression data of *Arapaima* sp. juveniles.** Relative expression data (raw values and means±S.D.) of seven digestive genes — *amy*, *plA2*, *lpl*, *try*, *ctr*, *pga*, and *pyy* — in the stomach, anterior intestine, middle intestine, posterior intestine, and liver of *Arapaima* sp. juveniles (3.19±0.03 cm TL), and across five developmental stages (D1 to D15, 3.19±0.03 to 5.05±0.34 cm TL). Data correspond to results presented in Figs 11 and 12.
(XLSX)

## Acknowledgments

This work was conducted in the framework of the network LARVAplus 'Strategies for the development and improvement of fish larvae production in Ibero-America' (117RT0521, Ibero-American Program of Science and Technology for Development, CYTED, Spain).

## Author contributions

**Conceptualization:** Maria J Darias.

**Data curation:** Maria J Darias.

**Formal analysis:** Maria J Darias.

**Funding acquisition:** Maria J Darias.

**Investigation:** Maria J Darias, Guillain Estivals, Karl B. Andree, Christian Fernández-Méndez, Roger Bazán, Chantal Cahu, Diana Castro-Ruiz.

**Methodology:** Maria J Darias.

**Project administration:** Maria J Darias, Roger Bazán.

**Resources:** Maria J Darias, Karl B. Andree, Roger Bazán, Chantal Cahu, Enric Gisbert.

**Supervision:** Maria J Darias, Karl B. Andree, Enric Gisbert.

**Validation:** Maria J Darias.

**Visualization:** Maria J Darias.

**Writing – original draft:** Maria J Darias.

**Writing – review & editing:** Guillain Estivals, Karl B. Andree, Christian Fernández-Méndez, Chantal Cahu, Enric Gisbert, Diana Castro-Ruiz.

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
