## [Decision Letter · Decision Letter 0]

5 Nov 2024

PONE-D-24-43490Histological and molecular characterization of the digestive system of early weaned juveniles of Arapaima sp. reared in a recirculating aquaculture systemPLOS ONE

Dear Dr. Darias,

Thank you for submitting your manuscript to PLOS ONE. After careful consideration, we feel that it has merit but does not fully meet PLOS ONE’s publication criteria as it currently stands. Therefore, we invite you to submit a revised version of the manuscript that addresses the points raised during the review process.

We look forward to receiving your revised manuscript.

Kind regards,

Mohammed Fouad El Basuini, Professor

Academic Editor

PLOS ONE

Journal Requirements: When submitting your revision, we need you to address these additional requirements. 1. Please ensure that your manuscript meets PLOS ONE's style requirements, including those for file naming. The PLOS ONE style templates can be found at https://journals.plos.org/plosone/s/file?id=wjVg/PLOSOne_formatting_sample_main_body.pdf and https://journals.plos.org/plosone/s/file?id=ba62/PLOSOne_formatting_sample_title_authors_affiliations.pdf 2. Thank you for stating the following in the Acknowledgments Section of your manuscript: "This work was conducted in the framework of the network LARVAplus ‘Strategies for the development and improvement of fish larvae production in Ibero-America’ (117RT0521) funded by the Ibero-American Program of Science and Technology for Development (CYTED, Spain). D.C.-R. benefited from a travel grant from the National  Fund for Scientific, Technological Development, and Technological Innovation (FONDECYT, Peru) and a South-North mobilization grant from the IRD (France)."  We note that you have provided funding information that is not currently declared in your Funding Statement. However, funding information should not appear in the Acknowledgments section or other areas of your manuscript. We will only publish funding information present in the Funding Statement section of the online submission form. Please remove any funding-related text from the manuscript and let us know how you would like to update your Funding Statement. Currently, your Funding Statement reads as follows: "This research was funded by the Peruvian Project 192-FINCyT-IA-201 (Programa Nacional de Innovación para la Competitividad y Productividad, Innóvate-Perú, Peru) and the International Joint Laboratory ‘Evolution and Domestication of the Amazonian Ichthyofauna’ (LMI EDIA, IRD-IIAP-UAGRM, France, Peru, and Bolivia). The funders had no role in study design, data collection and analysis, decision to publish, or preparation of the manuscript." Please include your amended statements within your cover letter; we will change the online submission form on your behalf. 3. "Please include captions for your Supporting Information files at the end of your manuscript, and update any in-text citations to match accordingly. Please see our Supporting Information guidelines for more information: http://journals.plos.org/plosone/s/supporting-information.

Reviewers' comments:

Reviewer's Responses to Questions

**Comments to the Author**

1. Is the manuscript technically sound, and do the data support the conclusions?

Reviewer #1: Partly

Reviewer #2: Yes

2. Has the statistical analysis been performed appropriately and rigorously? 

Reviewer #1: N/A

Reviewer #2: Yes

3. Have the authors made all data underlying the findings in their manuscript fully available?

Reviewer #1: Yes

Reviewer #2: Yes

4. Is the manuscript presented in an intelligible fashion and written in standard English?

Reviewer #1: Yes

Reviewer #2: Yes

5. Review Comments to the Author

Reviewer #1: There are no recommendations

The most descriptive methodologies are without any references --why/how

There are no declaration of conflict interests

There are no authors contributions

There are no data availability

There is no ethical approval statement

Materials and methods very long and repeated --for example---LN/173---after the authors mentioned the histological analysis back to write it and start to mention again

There are no reference for the statistical analysis ?

LN/258--write as statistical analysis not else with an updated references

Results and discussion --going to be chapter like

Abstract:

There are no highlights

There are no graphical abstract

The keywords should be written after the abstract not before

LN/24-25--clarify the differences between the monotypic genus and multiple species

LN/26---tendencies---detailed this

LN/33-36--more details are requested

There are many things related to the introduction and materials and methods ---should not included at the abstract

Divide the abstract into backgrounds-aims/methods/results and conclusion

Add carnivores /aquaculture industry / food quality , histology and molecular analysis to the keywords

Introduction:

LN/50---determine the time to reach

LN/51--why R is capital with river

LN/58---what is this ?

LN/59-60---2 different style in writing the references

Introduction is extremely very long , repeated without any need

Aims need to be more clarified

Novelty of this study should be more highlighted

Materials and methods

LN/64--write as materials and methods not VS

LN/66--prepared--means what and add reference

LN/69-87---add reference

LN/73---why empty

LN/88-98--also without any references

LN/99-106--add reference

The authors did not mention any thing about the statistical analysis

Materials and methods :

What about the IACUC code --should be enclosed

LN/115-123---add reference

LN/124--write as a separate item ---experimental design

How many fishes did you use/groups

LN/135-143----very long --no need

LN/155---you should also write as a separate items--as histological analysis

Regarding the histological analysis ---complete details about the paraffin embedding technique should be enclosed with an updated references

Results :

Huge number of abbreviations were used---tabulate all as general

Write as Table(1):---------/Fig.(1):----etc--apply for all

There are no figure descriptions

Results ---

Is very long

What about the clinical signs of the treated fishes

What about the PM changes

What about the percentage of mortalities--if there is

Resulst and discussion --going to be chapter like

Discussion :

Rewrite it again

Conclusion :

Less than enough

References:

Some cited references need to be more update

Huge number of references were used (102)???

There are no plan for the study area ?

Reviewer #2: Comments to the Authors

I have reviewed your manuscript titled "Histological and molecular characterization of the digestive system of early weaned juveniles of Arapaima sp. reared in a recirculating aquaculture system" and would like to offer constructive feedback. I appreciate your effort and dedication and look forward to helping you enhance the quality of your work. Below, I will outline several points that require further consideration and improvement.

1. Abstract, Lines 23-29: Please delete or summarize the background information and proceed directly to the aim of the study.

2. Abstract, Lines 30-35: Please merge the aim of the study to avoid repetition.

3. Abstract: Please include specific recommendations in the conclusion to enhance clarity.

4. Abstract: In conclusion, please discuss your results and suggest directions for future studies.

5. Could you please provide the keywords for this study?

6. Introduction, Lines 96-112: The introduction is overly lengthy, and the aim is expressed in too many words. Please revise to clearly emphasize the aim of the study.

7. Lines 129 and Table 1: The value of crude lipid differs between the text and Table 1. Please verify and correct this discrepancy.

8. Did you use any anesthetics in the experiment?

9. Table 1: All full names of abbreviations, such as gene names, MUST be defined in the footnotes.

10. Line 261: Please italicize “(P < 0.05)”. Additionally, check the entire manuscript for consistent formatting.

11. I have noticed a few minor errors and spelling mistakes scattered throughout the manuscript. To enhance the linguistic quality of the manuscript, I would recommend considering professional language revision.

6. PLOS authors have the option to publish the peer review history of their article (what does this mean? ). If published, this will include your full peer review and any attached files.

**Do you want your identity to be public for this peer review?** For information about this choice, including consent withdrawal, please see our Privacy Policy .

Reviewer #1: **Yes: ** Professor/Elsayed Eldeeb Mehana

Reviewer #2: No

---

## [Author Response · Author response to Decision Letter 1]

30 Mar 2025

Replies to reviewer comments

We sincerely appreciate the time and effort of both reviewers in evaluation our manuscript. Their constructive feedback has helped us improve the clarity and quality of our study. Below, we provide detailed responses to each comment and outline the revisions made in the manuscript accordingly.

Reviewer #1: There are no recommendations

The most descriptive methodologies are without any references --why/how

Response: We have added additional references where necessary to support the methodological descriptions. However, some methods are widely used, while others follow manufacture’s protocols (e.g., molecular biology techniques), which do not usually require specific citations.

There are no declaration of conflict interests

Response: We are unsure which information is visible to the reviewers, but a competing interests’ statement was provided during submission, following the instructions of the journal. The authors have declared that no competing interests exist.

There are no authors contributions

Response: Author contributions were also included in the online submission in accordance with journal guidelines.

There are no data availability

Response: The required data availability statement was also provided during submission, indicating that all relevant data are included within the manuscript and its Supporting Information files.

There is no ethical approval statement

Response: This information was included in the original manuscript (L168-171; L146-149 in the revised version) and provided online during submission. At the time of the study, no local ethical committee existed at the IIAP in Iquitos, Peru, the country’s main Amazonian aquaculture research institution. However, as part of a Peruvian-French-Spanish collaboration, all animal procedures adhered to the European Union Council guidelines (2010/63/EU). Compliance was ensured by the lead author, who is certified in animal experimentation by the French Ministry of Agriculture and Fisheries.

Materials and methods very long and repeated --for example---LN/173---after the authors mentioned the histological analysis back to write it and start to mention again

Response: We reviewed and reorganized the manuscript to eliminate unintended repetition. The “Sampling and growth measurements” section originally described sample preservation methods for the subsequent histological and gene expression analyses (LL155-160 of the original submission), while the “Histological analyses” section detailed the specific analytical techniques. However, we have revised these sections for improved clarity.

There are no reference for the statistical analysis?

Response: We have added a reference.

LN/258—write as statistical analysis not else with an updated references

Response: The section title has been updated to “Statistical analyses,” as suggested.

Results and discussion --going to be chapter like

Response: The meaning of this comment is unclear. However, we have carefully revised both sections and have also shorten the discussion considerably.

Abstract:

There are no highlights

Response: PLOS ONE guidelines do not require highlights.

There are no graphical abstract

Response: Similarly, PLOS ONE guidelines do not request a graphical abstract.

The keywords should be written after the abstract not before

Response: Keywords were provided during submission and not in the manuscript, as per journal requirements. Their placement in the reviewer’s version may depend on how the submission system displays them.

LN/24-25--clarify the differences between the monotypic genus and multiple species

Response: A monotypic genus contains a single species, unlike other genera that contain multiple species. This is standard terminology that typically does not require further explanation.

LN/26---tendencies---detailed this

Response: This tendency refers to plant ingestion. However, this information is no longer present in the revised abstract.

LN/33-36--more details are requested

Response: We revised the abstract to include additional detail within the 300-word limit.

There are many things related to the introduction and materials and methods ---should not included at the abstract

Response: The abstract has been revised based on the reviewer’s feedback.

Divide the abstract into backgrounds-aims/methods/results and conclusion

Response: PLOS ONE guidelines (PLOS ONE style templates) do not require sectioned abstracts. We have structured the abstract according to their format: “The Abstract should: i) Describe the main objective(s) of the study; ii) Explain how the study was done, including any model organisms used, without methodological detail; iii) Summarize the most important results and their significance; iv) Not exceed 300 words.”

Add carnivores /aquaculture industry / food quality , histology and molecular analysis to the keywords

Response: It is generally recommended that keywords avoid duplicating words already used in the title. Therefore, we don’t believe that “histology and molecular analysis” should be added to the keywords. We have replaced “early weaning” with “Peruvian aquaculture”, to align with the reviewer’s recommendation (aquaculture industry) while maintaining a geographical focus. For the rest, we opted to retain our original keywords, as they accurately reflect the study’s focus and complement the title, such as “paiche” (common name of the species in Peru), “Amazon basin” (natural habitat), “digestive gene expression,” and “ontogeny.” Furthermore, this paper does not focus on “carnivores” (we are actually showing that our species has characteristics of an omnivorous species) or “food quality”.

Introduction:

LN/50---determine the time to reach

Response: These values are historical maximums observed in nature, which are less frequently observed today due to overfishing. The time necessary to achieve them is not well-documented, although it is thought to be between 4 and 7 years. However, we prefer not to include this information as it is not well-supported. Many papers on Arapaima spp. mention these historical data in a similar way to us.

LN/51--why R is capital with river

Response: “Essequibo River” is often capitalized as a formal name, but we have changed it to lowercase.

LN/58---what is this ?

Response: We have defined CITES as the Convention on International Trade in Endangered Species of Wild Fauna and Flora.

LN/59-60---2 different style in writing the references

Response: Schinz 1822 is a species authority notation, not a reference. Parentheses indicate reclassification under a different genus than originally described.

Introduction is extremely very long , repeated without any need

Response: The introduction has been streamlined to improve clarity and conciseness.

Aims need to be more clarified

Response: The aims have been reformulated to address the comments of both reviewers. The second reviewer requested a more synthesized description of the aims.

Novelty of this study should be more highlighted

Response: We have emphasized the study’s novelty in the introduction.

Materials and methods

Overall comment: The line numbers indicated by the reviewer in the next six comments do not correspond to the lines of the Material and Methods section. Where possible, we have identified and addressed the comments accordingly.

LN/64--write as materials and methods not VS

Response: We were unable to identify what this refers to.

LN/66--prepared--means what and add reference

Response: The term “prepared” is only used once in the paper in the L 176 of the original submission (“Paraffin blocks were prepared in an AP280-2Myr station”). We are not sure what the reviewer requires. Paraffin blocks are prepared by embedding the samples in paraffin, as described in the previous sentence. This is a standard procedure in sample preparation for histological analysis. The reference “Sarasquete and Gutiérrez, 2005”, which was inadvertently not formatted correctly, was already included and we have added an additional one.

LN/69-87---add reference

Response: If the previous comment regarding L 66 actually refers to L 176 in the initial submitted version, we assume that the reviewer may be referring to LL 179-197. If that is the case, we have added a reference in LL 159 and 174 to complement those that were already included in LL 162, 164, and 177.

LN/73---why empty

Response: We could not locate the term “empty” in the manuscript.

LN/88-98--also without any references

Response: If the reviewer refers to the “Partial mRNA amplification and identification” section, we have added reference (LL 198 and 206). However, we would like to highlight that, as mentioned in LL 184, 188, and 197, this section often follows manufacture’s protocols.

LN/99-106--add reference

Response: If the reviewer refers to the “Partial mRNA amplifcation and identification” section, references have been included in LL 227 and 233.

The authors did not mention any thing about the statistical analysis

Response: We are not sure what the reviewer refers to. The statistical analyses were described in the original LL 258-263. However, we have made several modifications (LL 242-249).

Materials and methods :

What about the IACUC code --should be enclosed

Response: As previously mentioned, animal experimental procedures are documented in LL146-149 and were also included in the submission process.

LN/115-123---add reference

Response: We have added relevant references in L 104 regarding the taxonomic uncertainties (which were already provided in the introduction). However, for the rest of the text, we cannot add references as it refers to our specific rearing conditions.

LN/124--write as a separate item ---experimental design

Response: Given that the description of our experimental design is concise, we believe presenting the information in a single subsection is more appropriate.

How many fishes did you use/groups

Response: The manuscript provides details on fish density and the water volume per tank. Specifically, 1 ind L-1 in 30 L = 30 fish per tank (LL 106-108).

LN/135-143----very long --no need

Response: Diet composition details are essential in aquaculture nutrition for reproducibility and comparisons with similar studies. Therefore, we cannot remove this information.

LN/155---you should also write as a separate items--as histological analysis

Response: We have integrated this text into the histological analysis section (LL 152-177), although it referred to sample preservation during experimentation rather than the analytical methods themselves.

Regarding the histological analysis ---complete details about the paraffin embedding technique should be enclosed with an updated references

Response: We have added some additional details; however, paraffin embedding is a very standard procedure that typically does not require further details or updated references.

Results :

Huge number of abbreviations were used---tabulate all as general

Response: The only abbreviations used in the main text of the results are TL (total length) and WW (wet weight), which are defined upon their first mention in the Introduction and Material and Methods sections, respectively (L 60 and L136), as well as gene name abbreviations, which have also been defined previously. It seems the reviewer may be confusing the figure captions appearing directly after the paragraph in which they are first cited (see comments below), where each abbreviation is defined in the respective figure caption.

Write as Table(1):---------/Fig.(1):----etc--apply for all

Response: We followed the guidelines mentioned in the PLOS ONE style templates (PLOSOne_formatting_sample_main_body.pdf), which specify citing figures as “Fig 1”, “Fig 2”, etc., and tables as “Table 1”, “Table 2”, etc.

There are no figure descriptions

Response: In accordance with PLOS ONE guidelines, the captions for the 12 figures appear directly after the paragraph in which they are first cited. It seems the reviewer may have mistaken them as part of the main text. Each of the paragraphs starting with Fig X. correspond to a figure caption:

Fig 1: LL 261-265

Fig 2: LL 275-280

Fig 3: LL 299-306

Fig 4: LL 308-316

Fig 5: LL 339-352

Fig 6: LL 378-396

Fig 7: LL 398-407

Fig 8: LL 459-468

Fig 9: LL 470-483

Fig 10: LL 506-512

Fig 11: LL 538-543

Fig 12: LL 561-565

Results ---

Is very long

Response: The level of detail in the main text is standard for studies describing the histology of the digestive system in fish species. However, since the reviewer may not have recognized the presence of figure captions, they might be assessing the results section as a single text block, including both the main text and figure captions. These captions are positioned throughout the results section after each relevant paragraph (see lines above), in accordance with the journal’s guidelines. Nearly 40% of the section consists of captions for the 12 figures, so the main text of the results section is considerably shorter than what the reviewer may have perceived.

What about the clinical signs of the treated fishes

Response: This is a classical nutritional study with a single diet and, as such, it did not involve fish treatments. Clinical signs were not a relevant variable.

What about the PM changes

Response: If “PM” refers to postmortem changes, these were not part of our study’s scope nor relevant to its conclusions.

What about the percentage of mortalities–if there is

Response: The survival rate is provided in L 259, from which the mortality rate can be easily inferred.

Resulst and discussion –going to be chapter like

Response: The discussion section has been rewritten to improve clarity and conciseness.

Discussion :

Rewrite it again

Response: The discussion section has been revised to reduce its length and improve the flow and clarity.

Conclusion :

Less than enough

Response: The conclusion sections has been expanded to provide a clearer summary of the findings.

References:

Some cited references need to be more update

Response: Older references are cited to credit foundational studies. We have reviewed all citations to confirm accuracy and relevance.

Huge number of references were used (102)???

Response: The number of references reflects the detail of our study, aligning with PLOS ONE’s policy to include any relevant works. However, the number of references has been reduced to 77 after rewriting the discussion section.

There are no plan for the study area ?

Response: Since this is a controlled experiment in an indoor recirculating aquaculture system located in a research institute, there is no geographic study area to map. Instead, the location is provided in the Materials and Methods section in LL 102-103.

Reviewer #2: Comments to the Authors

I have reviewed your manuscript titled "Histological and molecular characterization of the digestive system of early weaned juveniles of Arapaima sp. reared in a recirculating aquaculture system" and would like to offer constructive feedback. I appreciate your effort and dedication and look forward to helping you enhance the quality of your work. Below, I will outline several points that require further consideration and improvement.

1. Abstract, Lines 23-29: Please delete or summarize the background information and proceed directly to the aim of the study.

Response: We have revised the abstract following the reviewer’s comments.

2. Abstract, Lines 30-35: Please merge the aim of the study to avoid repetition.

Response: Done

3. Abstract: Please include specific recommendations in the conclusion to enhance clarity.

Response: Done. Thank you for this suggestion.

4. Abstract: In conclusion, please discuss your results and suggest directions for future studies.

Response: Done. Thank you for this suggestion.

5. Could you please provide the keywords for this study?

Response: Keywords were provided during the original submission and not included in the manuscript, as per journal requirements. Reviewer #1 commented on the keywords, so we understand that they are visible to the reviewers. The keywords are: Paiche; Amazon basin; Peruvian aquaculture; digestive gene expres

---

## [Editor Report · Decision Letter 1]

1 Apr 2025

Histological and molecular characterization of the digestive system of early weaned juveniles of Arapaima sp. reared in a recirculating aquaculture system

PONE-D-24-43490R1

Dear Dr. Darias,

We’re pleased to inform you that your manuscript has been judged scientifically suitable for publication and will be formally accepted for publication once it meets all outstanding technical requirements.

Kind regards,

Mohammed Fouad El Basuini, Professor

Academic Editor

PLOS ONE
---

## [Editor Report · Acceptance letter]

PONE-D-24-43490R1

PLOS ONE

Dear Dr. Darias,

I'm pleased to inform you that your manuscript has been deemed suitable for publication in PLOS ONE. Congratulations! Your manuscript is now being handed over to our production team.

Kind regards,

on behalf of

Prof. Mohammed Fouad El Basuini

Academic Editor

PLOS ONE